# Do We Need Domain-Specific Embedding Models? An Empirical Investigation on Finance Domain and a Domain-MTEB Benchmark

## Abstract

Embedding models play a crucial role in representing and retrieving information across various NLP applications. Recent advancements in Large Language Models (LLMs) have further enhanced the performance of embedding models, which are trained on massive amounts of text covering almost every domain. These models are often benchmarked on general-purpose datasets like Massive Text Embedding Benchmark (MTEB), where they demonstrate superior performance. However, a critical question arises: Is the development of domain-specific embedding models necessary when general-purpose models are trained on vast corpora that already include specialized domain texts? In this paper, we empirically investigate this question, choosing the finance domain as an example. We introduce the Finance Massive Text Embedding Benchmark (FinMTEB), a counterpart to MTEB that consists of financial domain-specific text datasets. We evaluate the performance of seven state-of-the-art embedding models on FinMTEB and observe a significant performance drop compared to their performance on MTEB. To account for the possibility that this drop is driven by FinMTEB's higher complexity, we propose four measures to quantify dataset complexity and control for this factor in our analysis. Our analysis provides compelling evidence that state-of-the-art embedding models struggle to capture domain-specific linguistic and semantic patterns. Moreover, we find that the performance of general-purpose embedding models on MTEB is not correlated with their performance on FinMTEB, indicating the need for domain-specific embedding benchmarks for domain-specific embedding models. This study sheds light on developing domain-specific embedding models in the LLM era.

## 1 Introduction

Embedding models, which transform text sequences into dense vector representations, play a crucial role in various natural language processing (NLP) tasks (Mikolov et al., 2013; Pennington et al., 2014; Peters et al., 2018). The quality of text embeddings directly impacts the effectiveness of information retrieval, semantic understanding, and other downstream applications. Recently, many state-of-the-art embedding models have been developed using large language models (LLMs) as the foundational model (Wang et al., 2023; Li et al., 2023; Meng et al., 2024). Since LLMs are trained on massive text corpora spanning nearly every domain, these LLM-based embedding models have demonstrated superior and robust performance in general-purpose embedding benchmarks such as Massive Text Embedding Benchmark (MTEB) (Muennighoff et al., 2022).

Given that general-purpose embedding models are becoming the backbone of NLP applications, and companies like OpenAI and Cohere are offering general-purpose embeddings that potentially serve a wide range of industry applications, a critical question arises: *Do we still need domain-specific embedding models?* The answer is not immediately clear. On the one hand, as mentioned earlier, state-of-the-art embedding models are primarily built from general-purpose LLMs that have been trained on vast text corpora covering nearly every domain. There is no strong evidence suggesting that these models cannot grasp domain-specific languages or linguistic patterns. On the other hand, while there has been limited development of domain-specific embedding models, researchers have advocated for training domain-specific LLMs (Gururangan et al., 2020) to better

capture domain-specific terminology and semantics. For example, domain-specific LLMs such as BioMedLM (Bolton et al., 2024) for the biomedical domain, SaulLM-7B (Colombo et al., 2024) for the legal domain, and BloombergGPT (Wu et al., 2023) for the finance domain are pre-trained on large, domain specialized corpora.

To address this question, we empirically evaluate the necessity of domain-specific embedding models, focusing on the finance domain as our research context. We select the finance domain because financial NLP is a critical area within the research community, with a wealth of established financial NLP datasets (FiQA, 2018; Islam et al., 2023; Liu et al., 2024a; Ju et al., 2023; Sinha & Khandait, 2021; Mukherjee et al., 2022; Malo et al., 2014). The importance of representing financial texts in downstream applications has also driven the development of finance-specific LLMs, such as BloombergGPT (Wu et al., 2023) and InvestLM (Yang et al., 2023b). Moreover, the complexity and specificity of the financial domain provide a unique opportunity to assess how effectively general-purpose embedding models can represent specialized texts.

We first develop the Finance Massive Text Embedding Benchmark (FinMTEB), a finance-specific counterpart to MTEB. FinMTEB consists of 64 datasets and, like MTEB, covers seven distinct tasks, including classification, clustering, retrieval, pair-classification, reranking, and semantic textual similarity. Unlike MTEB, all datasets in FinMTEB are based on financial text data, which feature substantially longer text sequences and token lengths. Using seven state-of-the-art embedding models, we observe a significant performance drop on the FinMTEB compared to the general-purpose MTEB. ANOVA analysis further indicates that this average performance drop is primarily driven by the differences between the benchmarks, rather than model-specific factors.

While the performance drop on FinMTEB may seem expected given the domain shift, one concern is whether the datasets in FinMTEB are inherently more complex than those in MTEB. Is the reduced performance a result of the benchmark's complexity, or do these models lack the necessary understanding of domain-specific context? If the FinMTEB datasets were of equal complexity to MTEB, we might not observe the same performance gap, suggesting that dataset complexity could be contributing to the performance decline.

To eliminate the confounding factor of dataset complexity, we propose four different measures to quantify complexity: ChatGPT's response error rate, dataset readability, information entropy, and text dependency distance. Our analysis shows that even when controlling for complexity, general-purpose embedding models still perform worse on domain-specific texts. Moreover, the more complex the domain-specific data, the greater the performance drop—although this trend is less prominent in general-purpose tasks on the MTEB. Collectively, this evidence suggests that state-of-the-art, general-purpose embedding models may not fully capture the linguistic nuances and semantic patterns unique to a particular domain.

Moreover, we observe that the performance of general-purpose embedding models on MTEB does not correlate with their performance on FinMTEB. Models that perform exceptionally well on general embedding tasks do not necessarily maintain their superiority in the financial domain. This underscores the importance of evaluating embedding models within the specific context in which they will be applied and emphasizes the necessity of domain-specific embedding benchmarks.

Our contributions in this paper are threefold:

- Our main research contribution is the empirical investigation into the necessity of domain-specific embedding models. To the best of our knowledge, this is one of the first studies to address the critical question of whether domain-specific embeddings are required, especially given the widespread adoption of general-purpose embedding models across various industry applications.

- Our analysis on the necessity of domain-specific embedding models is based on a rigorous evaluation framework. Rather than simply developing a domain benchmark and demonstrating a performance drop, our analysis accounts for dataset complexity, eliminating potential confounding factors. This allows us to conclude that the performance gap is due to the models' inability to encode domain-specific text, rather than inherent dataset complexity.

- The development of the FinMTEB dataset, as a byproduct of our study, may serve as a valuable resource for researchers and practitioners interested in building financial domain-specific embedding models.

## 2 RELATED WORK

### 2.1 GENERAL-PURPOSE EMBEDDING MODELS

Embedding models like Word2Vec (Mikolov et al., 2013) and GloVe (Pennington et al., 2014) lay the groundwork by capturing word-level semantics through contextual co-occurrence. The introduction of transformer-based models such as BERT (Devlin et al., 2019) and RoBERTa (Liu, 2019) marks a significant shift by utilizing deep bidirectional encoders, enabling contextualized word embeddings. Building on these, Sentence-BERT (Reimers & Gurevych, 2019) is designed to generate semantically meaningful sentence embeddings using Siamese and triplet networks, improving performance on semantic similarity tasks. Recent advancements in LLMs have driven the development of LLM-based embedding models, such as e5-mistral-7b-instruct (Wang et al., 2023) and gte-Qwen2-1.5B-instruct (Yang et al., 2024), which have achieved state-of-the-art performance across a wide range of NLP tasks.

### 2.2 DOMAIN-SPECIFIC MODELS

Different domains exhibit distinct linguistic patterns and terminologies, often requiring domain-specific models or adaptations for specialized tasks. Researchers have advocated for training domain-specific models or fine-tuning general models for particular domains (Gururangan et al., 2020). For instance, domain-specific LLMs like BioMedLM (Bolton et al., 2024) for biomedical text, SaulLM-7B (Colombo et al., 2024) for legal documents, and BloombergGPT (Wu et al., 2023) for financial applications are pre-trained on large domain-specific corpora. In addition, instruction-tuned domain-specific models such as InvestLM (Yang et al., 2023b) and FinGPT (Yang et al., 2023a) are fine-tuned for specific downstream tasks in the finance domain.

While domain-specific LLMs have been widely studied and developed, domain-specific embedding models have received relatively less attention. In the biomedical domain, models like BioWordVec (Zhang et al., 2019) and BioSentVec (Chen et al., 2019) generate word and sentence embeddings tailored to biomedical texts. In finance, FinBERT (Yang et al., 2020) is pre-trained on a large corpus of financial texts to enhance text encoding capabilities. However, most domain-specific embedding models are based on smaller language models instead of state-of-the-art LLMs. Since LLMs are trained on extensive data across multiple domains with numerous parameters, it remains unclear whether general-purpose LLM-based embeddings can adequately handle specialized texts. This paper aims to address this research gap.

### 2.3 EMBEDDING BENCHMARKS

To comprehensively evaluate embedding models, benchmarks like the Massive Text Embedding Benchmark (MTEB) (Muennighoff et al., 2022) have been established. MTEB assesses embedding models across a wide array of tasks using numerous datasets and languages. This extensive evaluation provides insights into a model's generalizability and effectiveness across different linguistic contexts and task types. Similarly, the BEIR benchmark (Thakur et al., 2021) focuses on the information retrieval task, encompassing 18 diverse datasets. While BEIR includes some domain-specific datasets such as FiQA (FiQA, 2018), it is not tailored for comprehensive domain analysis. The inclusion of a few specialized datasets does not fully address the unique challenges posed by domain-specific language and terminology. There are also scenario-specific RAG evaluation benchmarks like RAGeval (Zhu et al., 2024b). These benchmarks acknowledge the necessity for domain-specific evaluations, particularly highlighting the impact of accurate retrieval in specialized contexts. However, they primarily focus on retrieval tasks and often overlook other crucial embedding tasks such as semantic similarity and clustering.

162
163
164
165
166
167
168
169
170
171
172
173
174
175
176
177
178
179
180
181
182
183
184
185
186
187
188
189
190
191
192
193
194
195
196
197
198
199
200
201
202
203
204
205
206
207
208
209
210
211
212
213
214
215

## 2.4 DOMAIN-SPECIFIC MODEL BENCHMARKS

Numerous benchmarks tailored to specific domains have been developed with the emergence of domain-specific large language models (LLMs). For example, in the finance domain, benchmarks such as CFLUE (Zhu et al., 2024a), FinEval (Zhang et al., 2023), DocMath-Eval (Zhao et al., 2024), and FinanceBench (Islam et al., 2023) have been introduced to assess the comprehension capabilities of LLMs within financial contexts. Similarly, in the legal domain, LawBench (Fei et al., 2023) has been established to evaluate LLMs across a variety of legal tasks. Besides, MedBench (Liu et al., 2024b), MedEval (He et al., 2023), and DrBenchmark (Labrak et al., 2024) have been developed to test the proficiency in understanding and generating medical information. Most of these benchmarking papers conclude that general-purpose LLMs may fall short on domain tasks (Zhu et al., 2024a; Fei et al., 2023). The importance of domain adaptation has gradually gained attention (Ling et al., 2023). However, to our knowledge, there is little work benchmarking the embedding model's performance on domain texts.

## 3 THE FINMTEB BENCHMARK

In this section, we briefly introduce the proposed Finance MTEB (FinMTEB) benchmark, which serves as the foundation for our analysis. The construction of FinMTEB closely resembles the widely used general embedding benchmark, MTEB (Muennighoff et al., 2022).

### 3.1 FINMTEB TASKS

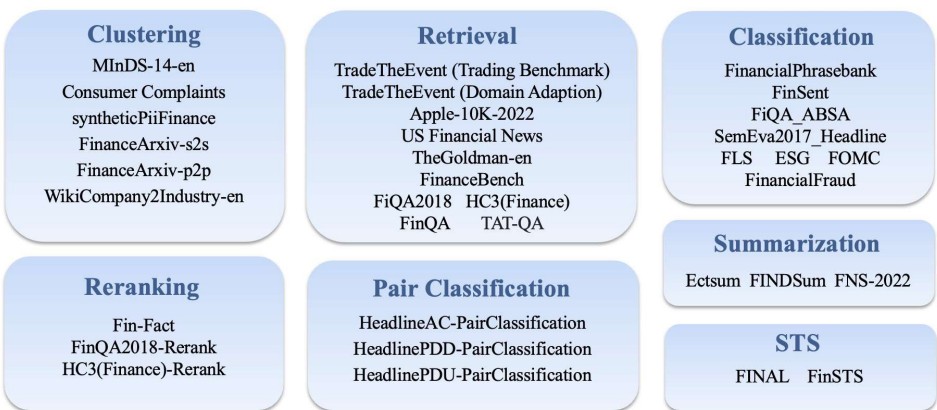

Figure 1: An overview of tasks and datasets used in FinMTEB. All the dataset descriptions and examples are provided in the Appendix D.

Figure 1 provides an overview of the tasks and datasets included in FinMTEB. Similar to MTEB (Muennighoff et al., 2022), FinMTEB includes seven embedding tasks, but with datasets specifically tailored to the finance domain, as follows.

**Semantic Textual Similarity (STS)** involves assessing the semantic similarity between two sentences from the financial text. For this task, we employ datasets such as FinSTS (Liu et al., 2024a) and FINAL (Ju et al., 2023) from company annual reports, along with other data types such as BQ-Corpus (Chen et al., 2018) sourced from the banking corpus.

**Retrieval** focuses on identifying the most relevant evidence in response to a query from a financial corpus. This task utilizes some popular finance QA datasets such as FinanceBench (Islam et al., 2023), FiQA2018 (FiQA, 2018) and HPC3 (Guo et al., 2023). These datasets pair each query with relevant contextual information. Additionally, we also develop specific queries for finance terms from various sources, such as the TradeTheEvent (Zhou et al., 2021), to further enhance the finance domain evaluation.

**Classification** involves predicting the label of a financial text based on its text embedding. The classification task includes multiple datasets, such as financial sentiment analysis (Malo et al., 2014;

FiQA, 2018; Cortis et al., 2017; Lu et al., 2023), Fed's monetary policy classification (Shah et al., 2023), and organization's strategy, as well as forward-looking statement classification (Yang et al., 2023b).

**Clustering** is the process of grouping sentences based on their embedding similarities. We compile a diverse and comprehensive corpus that includes consumer complaints from CFPB [1], financial papers from arXiv, company industry descriptions (Qader et al., 2018), financial events and intent detection(Gerz et al., 2021a).

**Reranking** includes a set of financial datasets that have the ranking of retrieved documents to user queries such as FinQA2018-Rerank (Chen et al., 2021).

**Pair-Classification** focuses on comparing the class label of two financial text. We use the data from AFQMC [2] and finance news headline (Sinha & Khandait, 2021).

**Summarization** focuses on summarizing the main content of the financial text. The corpus used for this task includes earnings call transcripts (Mukherjee et al., 2022), financial news (Lu et al., 2023), and Form 10-K filings (El-Haj et al., 2022).

In summary, FinMTEB consists of a total of 64 datasets, spanning seven different tasks. The key difference between MTEB and FinMTEB is that all datasets in FinMTEB are finance-domain specific, either previously used in financial NLP research or newly developed by the authors. Semantic similarity between datasets in FinMTEB are shown in Appendix A. The detailed dataset information and descriptions are presented in the Appendix D. The main scoring metric for each task is the same as used with that of the MTEB benchmark, and the details are presented in the Appendix E.

### 3.2 CHARACTERISTICS OF FINMTEB

Having built the FinMTEB benchmark, we now provide an analysis to understand its characteristics.

**Linguistic Pattern.** Table 1 presents a comparative analysis of linguistic features such as average sentence length, token length, syllables per token, and dependency distance (Oya, 2011) across two benchmarks. The results indicate that texts in FinMTEB consistently have longer and more complex sentences than those in MTEB, with an average sentence length of 26.37 tokens compared to 18.2 tokens in MTEB. This highlights significant linguistic differences between financial and general texts. However, does this difference warrant the need for a domain-specific embedding model? We will explore this question later.

Table 1: Comparison of Text Characteristics Between FinMTEB and MTEB. The numbers represent the average scores across all samples from all datasets.

| Benchmark | Sentence Length | Token Length | Syllables Per Token | Dependency Distance |
|---|---|---|---|---|
| MTEB | 18.20 | 4.89 | 1.49 | 2.49 |
| FinMTEB | 26.37 | 5.12 | 1.52 | 2.85 |

**Semantic Diversity.** We examine the inter-dataset semantic similarity. Using the all-MiniLM-L6-v2 model[3], we embed 1000 samples from each dataset, compute their averages to represent the dataset embedding, and measure inter-dataset similarity using cosine similarity. As shown in Appendix A, most datasets in FinMTEB have an inter-dataset similarity score below 0.6, with a mean cosine similarity of 0.4. Despite being finance-domain specific, this highlights the diverse narratives and contexts present in the financial datasets.

### 3.3 THE PERFORMANCE OF STATE-OF-THE-ART EMBEDDING MODELS ON FINMTEB

**General-purpose Embedding Models.** We consider **seven** state-of-the-art, general-purpose embedding models in our experiments. Specifically, we consider the following models: bge-en-icl (Xiao et al., 2023) and e5-mistral-7b-instruct (Wang et al., 2023), which are developed from Mistral-7B-v0.1 (Jiang et al., 2023); gte-Qwen2-1.5B-instruct (Li et al., 2023), developed from Qwen2

---

[1]https://huggingface.co/datasets/CFPB/consumer-finance-complaints

[2]https://tianchi.aliyun.com/dataset/106411

[3]https://huggingface.co/sentence-transformers/all-MiniLM-L6-v2

270
271
272
273

(Yang et al., 2024); bge-large-en-v1.5 (Xiao et al., 2023) and all-MiniLM-L12-v2 (Reimers & Gurevych, 2019), both developed from BERT (Devlin et al., 2019); instructor-base (Su et al., 2022) from T5Encoder (Raffel et al., 2020); and OpenAI's text-embedding-3-small (OpenAI, 2024b).

274
275
276
277
278

We evaluate the performance of these embedding models on FinMTEB tasks, with the results presented in Table 2, alongside their performance on MTEB for comparison. The results clearly demonstrate a significant performance drop on the FinMTEB benchmarks. For instance, the best-performing model on MTEB, bge-en-icl, achieves an average score of 71.67, while its performance on FinMTEB is notably lower, at 63.09.

279
280
281
282
283
284

Table 2: State-of-the-art Embedding Model Performance on MTEB and FinMTEB. The "MTEB Score" column represents performance on the MTEB benchmark (Muennighoff et al., 2022) as reported on the Hugging Face MTEB Leaderboard [4]. The "FinMTEB Score" column shows the average performance score evaluated on the proposed FinMTEB benchmark. To ensure a fair comparison, FinMTEB uses the same evaluation metrics as MTEB. More evaluation results on other SoTA models are presented in Appendix B.

285
286
287
288
289
290
291
292
293

| Embedding Model | Base Model | Embedding Dimensions | MTEB Score | FinMTEB Score |
|---|---|---|---|---|
| bge-en-icl | Mistral | 4096 | 71.67 | 63.09 ($\downarrow -8.58$) |
| gte-Qwen2-1.5B-instruct | Qwen2 | 1536 | 67.16 | 59.98 ($\downarrow -7.18$) |
| e5-mistral-7b-instruct | Mistral | 4096 | 66.63 | 64.75 ($\downarrow -1.88$) |
| bge-large-en-v1.5 | Bert | 1024 | 64.23 | 58.95 ($\downarrow -5.28$) |
| text-embedding-3-small | - | 1536 | 62.26 | 61.36 ($\downarrow -0.90$) |
| instructor-base | T5Encoder | 768 | 59.54 | 54.79 ($\downarrow -4.75$) |
| all-MiniLM-L12-v2 | Bert | 384 | 56.53 | 54.31 ($\downarrow -2.22$) |

294

### 3.3.1 What Drives the Performance Gap?

295
296
297
298
299
300

Having observed a performance discrepancy between general-purpose embedding models across the two benchmarks, we aim to investigate what drives this difference. Specifically, we consider two possible factors: (1) the **model** used (i.e., which embedding model is applied), and (2) the **domain** (i.e., whether it's the general benchmark, MTEB, or the domain-specific benchmark, FinMTEB). Since we evaluate seven embedding models across two domains, this results in 14 unique model-domain combinations.

301
302
303
304
305
306
307

To facilitate statistical analysis, we employ bootstrapping methods to generate a large sample dataset. For each task in both MTEB and FinMTEB, we aggregate the task's datasets into a task pool. From each task pool, we randomly sample 50 examples to form a bootstrap sample and evaluate the embedding model's performance on this bootstrap. We repeat this process 500 times, yielding 500 bootstraps for each combination. Thus, we have 14 unique combinations (model and domain), each with 500 bootstraps and corresponding performance scores.

308
309
310
311

Table 3: ANOVA analysis results. The reported numbers represent the partial eta squared (effect size) for each factor (Model or Domain). Asterisks indicate statistical significance levels, with $**$ denoting $p$-value $< 0.05$.

312
313
314

| | STS | Classification | Retrieval | Reranking | Clustering | Pair-classification | Summarization | Average |
|---|---|---|---|---|---|---|---|---|
| **Model** | 0.79** | 0.02** | 0.89** | 0.30** | 0.04** | 0.00 | 0.11** | 0.00 |
| **Domain** | 0.11** | 0.23** | 0.31** | 0.05** | 0.82** | 0.63** | 0.12** | 1.00** |

315
316
317
318
319
320
321
322

We present the ANOVA analysis results in Appendix C. First, the results indicate that the choice of embedding model (Model factor) significantly impacts performance in most tasks, such as STS (0.79), Retrieval (0.89), and Reranking (0.30), with the exception of Pair-classification (0.00), where model choice has no significant impact. Second, the Domain factor also shows significant effects across all embedding tasks. Interestingly, the average scores reveal that, from an overall perspective, the Model factor has little impact on performance, with an effect size of 0.00 and an insignificant $p$-value. This suggests that while individual models may excel at specific tasks, their performance discrepancies balance out when averaged. However, the Domain factor (1.00) demonstrates a much

323

---

[4]https://huggingface.co/spaces/mteb/leaderboard

more prominent influence, underscoring the necessity for domain-specific models or fine-tuning when addressing specialized tasks like those in finance.

**Research Question.** While the performance drop on FinMTEB and the subsequent ANOVA analysis suggests that domain-specific embedding tasks may pose greater challenges for general-purpose embedding models, does this necessarily indicate a need for domain-specific models? Not necessarily. The difference in datasets between FinMTEB and MTEB could contribute to the observed performance drop. For instance, FinMTEB datasets might be inherently more difficult or linguistically complex compared to those in MTEB as illustrated in Table 1. If both benchmarks contained datasets of equivalent complexity, general-purpose models might even perform better on FinMTEB tasks. Therefore, the performance drop does not necessarily imply that the models fail to understand domain-specific language or concepts. To draw meaningful conclusions about the necessity of domain-specific models, we must first control for differences in dataset difficulty. In the next section, we will analyze model performance while considering these inherent differences between the FinMTEB and MTEB datasets.

# 4 PERFORMANCE ANALYSIS AFTER CONTROLLING FOR DATASET COMPLEXITY

To answer the above research question, we conduct a detailed analysis of the embedding models' performance, while accounting for dataset complexity.

## 4.1 QUANTIFYING DATASET COMPLEXITY

We propose four different measures to quantify a dataset's complexity.

**ChatGPT Error Rate.** The first measure quantifies how challenging it is for ChatGPT to answer a dataset's questions. Specifically, for each example in the dataset across different tasks, we reformat the example into a question-and-answer pair, as shown in Appendix G, and prompt GPT-4o mini (OpenAI, 2024a). The rationale is that if ChatGPT fails to answer a question correctly, it indicates the difficulty level of the question. Additionally, since state-of-the-art LLM-based embedding models present each query with an instruction in a question-answer format, we use the ChatGPT error rate as an indicator of dataset complexity.

**Information Theory.** We borrow the concept of information entropy from information theory to measure the complexity of a text sequence. Information entropy is calculated as the average amount of information produced by a stochastic source of data. Higher Information Entropy indicates text that contains more information or is less predictable, potentially implying greater complexity.

**Readability.** We also use readability to measure dataset complexity, specifically applying the Gunning Fog Index (Gunning, 1952), which factors in sentence length and the number of complex words. The index estimates the years of formal education required to understand a text on the first reading. A higher Gunning Fog Index score indicates more complex sentences.

**Mean Dependency Distance.** Finally, we measure linguistic complexity using the dependency distance between two syntactically related words in a sentence (Oya, 2011). A longer dependency distance indicates that more context is needed for comprehension, reflecting greater sentence complexity.

For all of these four complexity measures, a higher score indicates higher dataset complexity. Details on the measures and their calculations are provided in Appendix F.

## 4.2 SUBGROUP ANALYSIS: EMBEDDING PERFORMANCE VS. DATASET COMPLEXITY

We conduct a subgroup analysis to examine the impact of the domain on embedding model performance. First, we calculate dataset complexity using one of the four complexity measures and categorize the datasets into three subgroups: low, medium, and high complexity. This ensures that METB and FinMTEB datasets within each subgroup have the same level of complexity. We then calculate the average performance score of seven LLM-based embedding models across datasets

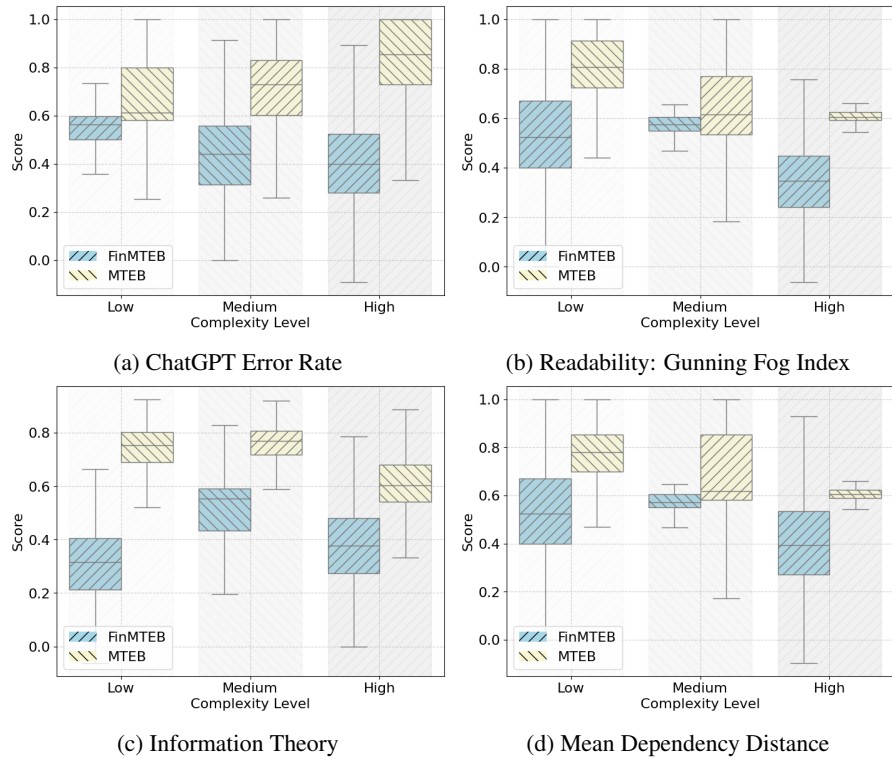

Figure 2: Subgroup analysis results. The x-axis represents the three dataset complexity levels: Low, Medium, and High. The y-axis reports the average score for each dataset across all benchmark tasks.

within each subgroup. The results of this analysis appear in Figure 4.2. The subgroup analysis reveals two key findings.

**First, embedding models perform substantially worse on FinMTEB datasets compared to MTEB datasets, even after accounting for dataset complexity.** The performance of embedding models on FinMTEB datasets is consistently lower than on MTEB datasets within the same group. This performance gap persists regardless of the complexity measure used. Given that datasets in the same subgroup have similar levels of complexity (e.g., comparable ChatGPT error rates), the lower performance on FinMTEB tasks suggests that state-of-the-art LLM-based embedding models struggle with encoding domain-specific terminologies and semantic patterns.

**Second, embedding models perform worst on FinMTEB datasets with the highest complexity levels.** For example, using readability as a complexity measure, the average performance of embedding models on high-complexity FinMTEB datasets is around 0.3, significantly lower than their performance on low-complexity datasets (around 0.5) and medium-complexity datasets (around 0.6). This further highlights the challenge embedding models face in capturing complex, domain-specific language and semantics.

### 4.3 REGRESSION ANALYSIS: EMBEDDING PERFORMANCE VS. DATASET COMPLEXITY

Table 4: Regression analysis results. The numbers represent the estimated coefficient values, with standard errors in parentheses. $**$ denotes significance at the $p < 0.05$ level.

| Domain | ChatGPT Error Rate | | Readability | | Information Theory | | Mean Dependency Distance | |
|---|---|---|---|---|---|---|---|---|
| | MTEB | FinMTEB | MTEB | FinMTEB | MTEB | FinMTEB | MTEB | FinMTEB |
| Intercept | 0.6619** | 0.6097** | 0.8739** | 0.6976** | 0.5905** | 0.4766** | 1.2341** | 0.9357** |
| | (0.003) | (0.003) | (0.002) | (0.005) | (0.002) | (0.002) | (0.007) | (0.008) |
| Complexity | -0.2193** | -0.4637** | -0.0278** | -0.0202** | 0.0168** | -0.0092** | -0.2703** | -0.1810** |
| | (0.009) | (0.010) | (0.000) | (0.000) | (0.001) | (0.001) | (0.003) | (0.003) |

Furthermore, we conduct a regression analysis to examine the relationship between dataset complexity and embedding model performance. Specifically, for each dataset complexity measure, we run two ordinary least squares (OLS) regression models—one for the MTEB datasets and one for the FinMTEB datasets. We normalize the variables Performance and Complexity to a range between 0 and 1 using min-max normalization. The regression specification is as follows:

$$\text{Performance} = \text{Intercept} + \beta \times \text{Complexity} + \epsilon \tag{1}$$

where $\beta$ is the coefficients of the covariate (i.e., dataset complexity), and $\epsilon$ is the error term.

The regression results are presented in Table 4 and are largely consistent with the findings from the subgroup analysis. First, there is a negative relationship between dataset complexity and embedding model performance, indicating that models significantly struggle with domain-specific texts exhibiting higher linguistic complexity. Second, the intercept for the MTEB datasets is consistently higher than that for FinMTEB. Given that the Complexity variable is normalized between 0 and 1, these results suggest a significant performance gap between embedding models on MTEB and FinMTEB, even when controlling for the same level of dataset complexity.

Overall, both the subgroup and regression analyses demonstrate that the performance drop reported in Table 2 is not driven by differences in dataset complexity between MTEB and FinMTEB benchmarks. Rather, it suggests that state-of-the-art, general-purpose embedding models may not fully capture the linguistic nuances and semantic patterns specific to a given domain.

## 5 DOMAIN-SPECIFIC EMBEDDING BENCHMARK IS NEEDED

Another key consideration when discussing domain-specific embeddings is whether we need a domain-specific embedding benchmark. While it may seem intuitive to say yes, there is little empirical evidence supporting this assumption. To explore this question, we evaluate the performance ranking of embedding models on both the general MTEB and FinMTEB datasets, calculating Spearman's rank correlation between the two. The results, shown in Table 5, indicate that the ranking correlation is not statistically significant (p-values all greater than 0.05). In other words, a general-purpose embedding model performing well on MTEB does not necessarily perform well on domain-specific tasks. This suggests the necessity of developing domain-specific embedding benchmarks for evaluating domain-specific embeddings. Therefore, the development of FinMTEB constitutes a significant contribution to benchmarking embedding models specifically for the financial domain.

Table 5: Spearman's correlation of embedding models' performance on MTEB and FinMTEB across different tasks. The p-value indicates that all correlations are statistically insignificant, suggesting a lack of evidence for a relationship between embedding model performance on the two benchmarks.

|  | STS | Classification | Retrieval | Reranking | Clustering | Pair-classification | Summarization |
|---|---|---|---|---|---|---|---|
| **Correlation** | 0.30 | -0.80 | 0.30 | -0.10 | -0.70 | -0.30 | 0.60 |
| **p-value** | 0.62 | 0.10 | 0.62 | 0.87 | 0.18 | 0.62 | 0.28 |

## 6 CONCLUSION

In this study, we empirically investigate a seemingly intuitive yet practically important question: do we need domain-specific embedding models? To rigorously address this, we use the finance domain as an example and develop the FinMTEB benchmark, which comprises a large variety of domain-specific (i.e., finance) embedding tasks. Additionally, we propose four methods to quantify dataset complexity. Our comparative analysis reveals that state-of-the-art LLM-based embedding models exhibit a substantial performance gap between general (MTEB) and domain-specific (FinMTEB) benchmarks. More importantly, this gap persists even when accounting for dataset complexity. The empirical results provide strong evidence that, despite being trained on vast amounts of data that likely include various domains, LLM-based embedding models still fall short in capturing domain-specific linguistic and semantic patterns. Given the widespread use of embedding models in information retrieval and semantic search applications, this highlights the need for further adaptation of these models to specific domains in order to improve their utility. Moreover, the development of the

FinMTEB benchmark can serve as a valuable resource for researchers and practitioners interested in financial-specific embedding models.

While this study presents compelling evidence for the necessity of domain-specific embedding models, the challenge of how to train these models remains. Should we adapt domain-specific embeddings from a domain-specific LLM, or should we develop domain-specific datasets and fine-tune a general-purpose LLM? We hope to see more research in this direction to further advance AI's capabilities in handling domain-specific tasks effectively.

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

## A  DATASET SEMANTIC SIMILARITY

Figure 3 presents the semantic similarity across all datasets in the Finance MTEB benchmark. The semantic similarity is calculated by cosine similarity.

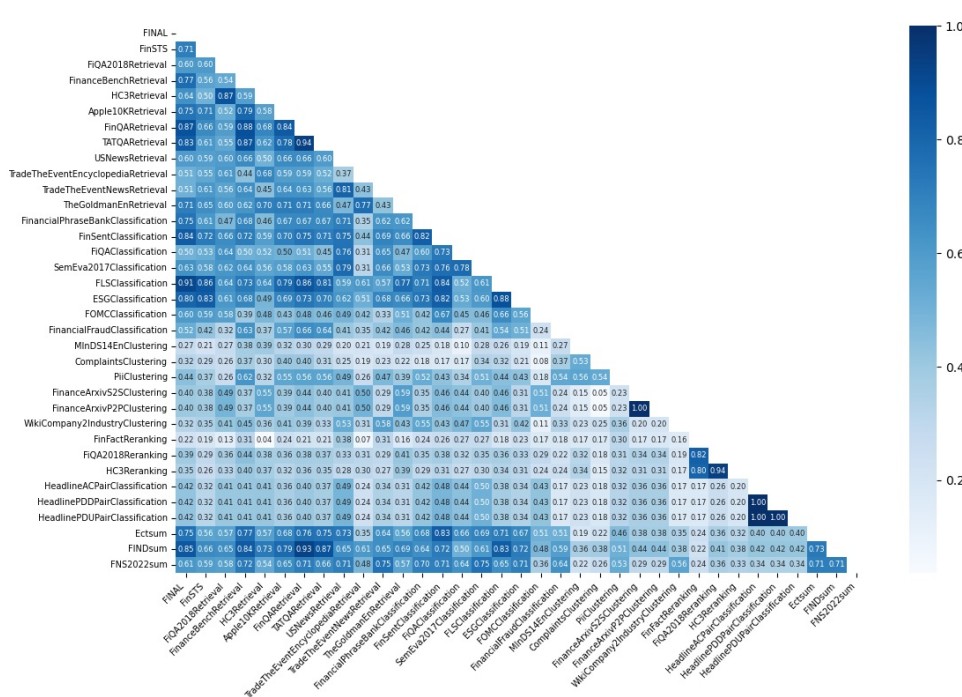

Figure 3: Semantic similarity across all the datasets in FinMTEB benchmark.

## B  ADDITIONAL STATE-OF-THE-ART EMBEDDING MODEL PERFORMANCE

## C  ANOVA DATA

Table 7 illustrates the full results of ANOVA analysis.

Table 6: Performance metrics of more state-of-the-art (SoTA) models on FinMTEB and their testing times (batch size = 512)

| Model | STS | Retr. | Rerank. | Class. | Summ. | PairClass. | Clus. | Avg. | Duration (/hr) |
|---|---|---|---|---|---|---|---|---|---|
| Echo Embedding (Springer et al., 2024) | 0.4380 | 0.6443 | 0.9765 | 0.6525 | 0.4722 | 0.6261 | 0.5776 | 0.6267 | 12.00 |
| AnglE-BERT (Li & Li, 2023) | 0.3080 | 0.5730 | 0.9650 | 0.6439 | 0.5049 | 0.6891 | 0.5774 | 0.6088 | 8.00 |
| NV-Embed v2 (Lee et al., 2024) | 0.3739 | 0.7061 | 0.9822 | 0.6393 | 0.5103 | 0.6043 | 0.6096 | 0.5739 | 5.60 |
| BeLLM (Li & Li, 2024) | 0.3919 | 0.0169 | 0.5661 | 0.1168 | 0.3906 | 0.5682 | 0.0685 | 0.3027 | 11.98 |

Table 7: **Analysis of Variance (ANOVA) Results Across Tasks and Factors.** *Factor* represents the independent variables analyzed: **Model Factor** pertains to variations attributed to different models, and **Domain Factor** pertains to variations due to different domains (MTEB or FinMTEB). **Residual** refers to the unexplained variance. The **Sum of Squares**, **Degrees of Freedom**, **F-Statistic**, and **p-value** are presented for each factor within each task. Asterisks denote significance levels, with lower p-values indicating higher statistical significance. The Domain Factor consistently shows high significance across all tasks.

| Task | Factor | Sum of Squares | Degrees of Freedom | F-Statistic | p-value |
|---|---|---|---|---|---|
| **Classification** | Model Factor | 4.17 | 6.00 | 25.55 | $3.41 \times 10^{-30}$ |
| | Domain Factor | 56.82 | 1.00 | 2086.30 | $\approx 0$ |
| | Residual | 190.42 | 6992.00 | NA | NA |
| **Retrieval** | Model Factor | 104.25 | 6.00 | 9052.57 | $\approx 0$ |
| | Domain Factor | 6.16 | 1.00 | 3207.72 | $\approx 0$ |
| | Residual | 13.42 | 6992.00 | NA | NA |
| **STS** | Model Factor | 10.55 | 6.00 | 149.00 | $1.64 \times 10^{-178}$ |
| | Domain Factor | 304.09 | 1.00 | 25761.71 | $\approx 0$ |
| | Residual | 82.53 | 6992.00 | NA | NA |
| **Clustering** | Model Factor | 0.29 | 6.00 | 47.60 | $1.59 \times 10^{-57}$ |
| | Domain Factor | 32.25 | 1.00 | 32161.37 | $\approx 0$ |
| | Residual | 7.01 | 6992.00 | NA | NA |
| **Summarization** | Model Factor | 12.98 | 6.00 | 145.31 | $2.90 \times 10^{-174}$ |
| | Domain Factor | 14.49 | 1.00 | 973.32 | $3.60 \times 10^{-200}$ |
| | Residual | 104.07 | 6992.00 | NA | NA |
| **Reranking** | Model Factor | 5.38 | 6.00 | 489.05 | $\approx 0$ |
| | Domain Factor | 0.64 | 1.00 | 346.78 | $1.39 \times 10^{-75}$ |
| | Residual | 12.84 | 7002.00 | NA | NA |
| **Pair Classification** | Model Factor | 0.25 | 6.00 | 1.97 | 0.07 |
| | Domain Factor | 249.19 | 1.00 | 11989.92 | $\approx 0$ |
| | Residual | 145.31 | 6992.00 | NA | NA |
| **Average** | Model Factor | 0.00 | 6.00 | 1.34 | 0.37 |
| | Domain Factor | 0.08 | 1.00 | 253.87 | $\approx 0$ |
| | Residual | 0.00 | 6.00 | NA | NA |

# D DATASETS

The detailed description of each dataset used in this work is listed in the Table tables 8 to 14.

Table 8: Summary of STS Datasets

| Dataset Name | Language | Description |
|---|---|---|
| FINAL (Ju et al., 2023) | English | A dataset designed for discovering financial signals in narrative financial reports. |
| FinSTS (Liu et al., 2024a) | English | A dataset focused on detecting subtle semantic shifts in financial narratives. |
| AFQMC [5] | Chinese | A Chinese dataset for customer service question matching in the financial domain. |
| BQ-Corpus (Chen et al., 2018) | Chinese | A large-scale Chinese corpus for sentence semantic equivalence identification (SSEI) in the banking domain. |

---

[5]https://tianchi.aliyun.com/dataset/106411

Table 9: Summary of Retrieval Datasets

| Dataset Name | Language | Description |
|---|---|---|
| FiQA2018 (FiQA, 2018) | English | Financial opinion mining and question answering dataset. |
| FinanceBench (Islam et al., 2023) | English | Open book financial question answering dataset. |
| HC3(Finance) (Guo et al., 2023) | English | A human-ChatGPT comparison corpus in the finance domain. |
| Apple-10K-2022 [6] | English | A retrieval-augmented generation (RAG) benchmark for finance applications. |
| FinQA (Chen et al., 2021) | English | Financial numerical reasoning dataset with structured and unstructured evidence. |
| TAT-QA (Zhu et al., 2021) | English | Question answering benchmark combining tabular and textual content in finance. |
| US Financial News [7] | English | Finance news articles paired with headlines and stock ticker symbols. |
| TradeTheEvent (Trading Benchmark) (Zhou et al., 2021) | English | Finance news articles paired with headlines and stock ticker symbols. |
| TradeTheEvent (Domain Adaption) (Zhou et al., 2021) | English | Financial terms and explanations dataset. |
| TheGoldman-en | English | English version of the Goldman Sachs Financial Dictionary. |
| FinTruthQA (Xu et al., 2024) | Chinese | Dataset for evaluating the quality of financial information disclosure. |
| Fin-Eva (Retrieval task) [8] | Chinese | Financial scenario QA dataset focusing on retrieval tasks. |
| AlphaFin (Li et al., 2024) | Chinese | Comprehensive financial dataset including NLI, QA, and stock trend predictions. |
| DISC-FinLLM (Retrieval Part Data) (Chen et al., 2023) | Chinese | Financial scenario QA dataset. |
| FinQA (from DuEE-fin) (Lu et al., 2023) | Chinese | Financial news bulletin event quiz dataset. |
| DISC-FinLLM (Computing) (Chen et al., 2023) | Chinese | Financial scenario QA dataset focusing on numerical tasks. |
| SmoothNLP [9] | Chinese | Chinese finance news dataset. |
| THUCNews (Sun et al., 2016) | Chinese | Chinese finance news dataset. |
| Fin-Eva (Terminology) [10] | Chinese | Financial terminology dataset used in the industry. |
| TheGoldman-cn | Chinese | Chinese version of the Goldman Sachs Financial Dictionary. |

Table 10: Summary of Classification Datasets

| Dataset Name | Language | Description |
| --- | --- | --- |
| FinancialPhrasebank (Malo et al., 2014) | English | Polar sentiment dataset of sentences from financial news, categorized by sentiment into positive, negative, or neutral. |
| FinSent (Yang et al., 2023b) | English | Polar sentiment dataset of sentences from the financial domain, categorized by sentiment into positive, negative, or neutral. |
| FiQA_ABSA (FiQA, 2018) | English | Polar sentiment dataset of sentences from the financial domain, categorized by sentiment into positive, negative, or neutral. |
| SemEva2017_Headline (Cortis et al., 2017) | English | Polar sentiment dataset of sentences from the financial domain, categorized by sentiment into positive, negative, or neutral. |
| FLS (Yang et al., 2023b) | English | A finance dataset detects whether the sentence is a forward-looking statement. |
| ESG (Yang et al., 2023b) | English | A finance dataset performs sentence classification under the environmental, social, and corporate governance (ESG) framework. |
| FOMC (Shah et al., 2023) | English | A task of hawkish-dovish classification in finance domain. |
| Financial-Fraud [11] | English | This dataset was used for research in detecting financial fraud. |
| FinNSP (Lu et al., 2023) | Chinese | Financial negative news and its subject determination dataset. |
| FinChina (Lan et al., 2023) | Chinese | Polar sentiment dataset of sentences from the financial domain, categorized by sentiment into positive, negative, or neutral. |
| FinFE (Lu et al., 2023) | Chinese | Financial social media text sentiment categorization dataset. |
| OpenFinData [12] | Chinese | Financial scenario QA dataset including sentiment task. |
| MDFEND-Weibo2 (finance) (Nan et al., 2021) | Chinese | Fake news detection in the finance domain. |

Table 11: Summary of Clustering Datasets

| Dataset Name | Language | Description |
| --- | --- | --- |
| MInDS-14-en (Gerz et al., 2021b) | English | MINDS-14 is a dataset for intent detection in e-banking, covering 14 intents across 14 languages. |
| Consumer Complaints (CFPB, 2024) | English | The Consumer Complaint Database is a collection of complaints about consumer financial products and services that sent to companies for response. |
| Synthetic PII finance (Watson et al., 2024) | English | Synthetic financial documents containing Personally Identifiable Information (PII). |
| FinanceArxiv-s2s [13] | English | Clustering of titles from arxiv (q-fin). |
| FinanceArxiv-p2p | English | Clustering of abstract from arxiv (q-fin). |
| WikiCompany2Industry-en [14] | English | Clustering the related industry domain according to the company description. |
| MInDS-14-zh (Gerz et al., 2021b) | Chinese | MINDS-14 is a dataset for intent detection in e-banking, covering 14 intents across 14 languages. |
| FinNL (Lu et al., 2023) | Chinese | Financial news categorization dataset. |
| CCKS2022 (CCKS, 2022) | Chinese | Clustering of financial events. |
| CCKS2020 | Chinese | Clustering of financial events. |
| CCKS2019 | Chinese | Clustering of financial events. |

Table 12: Summary of Summarization Datasets

| Dataset Name | Language | Description |
| --- | --- | --- |
| Ectsum (Mukherjee et al., 2022) | English | A Dataset For Bullet Point Summarization of Long Earnings Call Transcripts. |
| FINDSum (Liu et al., 2022) | English | A Large-Scale Dataset for Long Text and Multi-Table Summarization. |
| FNS-2022 (El-Haj et al., 2022) | English | Financial Narrative Summarisation for 10K. |
| FiNNA (Lu et al., 2023) | Chinese | A financial news summarization dataset. |
| Fin-Eva (Headline) | Chinese | A financial summarization dataset. |
| Fin-Eva (Abstract) | Chinese | A financial summarization dataset. |

---

[6]https://lighthouz.ai/blog/rag-benchmark-finance-apple-10K-2022/
[7]https://www.kaggle.com/datasets/jeet2016/us-financial-news-articles
[8]https://github.com/alipay/financial_evaluation_dataset/tree/main
[9]https://github.com/smoothnlp/SmoothNLP
[10]https://github.com/alipay/financial_evaluation_dataset/tree/main
[11]https://github.com/amitkedia007/Financial-Fraud-Detection-Using-LLMs/tree/main
[12]https://github.com/open-compass/OpenFinData?tab=readme-ov-file

Table 13: Summary of Reranking Datasets

| Dataset Name | Language | Description |
|---|---|---|
| Fin-Fact | English | A Benchmark Dataset for Financial Fact Checking and Explanation Generation. |
| FiQA2018 | English | Financial opinion mining and question answering. |
| HC3(Finance) | English | A human-ChatGPT comparison finance corpus. |
| Fin-Eva (Retrieval task) | Chinese | Financial scenario QA dataset including retrieval task. |
| DISC-FinLLM (Retrieval Part Data) | Chinese | Financial scenario QA dataset. |

Table 14: Summary of PairClassification Datasets

| Dataset Name | Language | Description |
|---|---|---|
| HeadlineAC-PairClassification (Sinha & Khandait, 2021) | English | Financial text sentiment categorization dataset. |
| HeadlinePDD-PairClassification (Sinha & Khandait, 2021) | English | Financial text sentiment categorization dataset. |
| HeadlinePDU-PairClassification (Sinha & Khandait, 2021) | English | Financial text sentiment categorization dataset. |
| AFQMC | Chinese | Ant Financial Question Matching Corpus. |

# E    MAIN METRIC

**Semantic Textual Similarity (STS):** The main metric used to measure performance in this task is Spearman's rank correlation of predicted cosine similarity scores with the true similarity score.

**Classification:** The main metric for evaluating is accuracy, ensuring that the model's assessment is based on different types of financial texts and frameworks.

**Clustering:** The main evaluation metric for this task is the V-measure (Rosenberg & Hirschberg, 2007), which assesses the quality of the clusters by examining both the completeness and the homogeneity of the data within each group.

**Rerank:** The main evaluation metric for reranking in Finance MTEB is Mean Average Precision (MAP).

**Pair-Classification:** The main evaluation metric for Pair-Classification is Average Precision (AP), which measures the model's accuracy across various decision thresholds.

**Summarization:** Summarization is evaluated based on the correlation between dense embeddings derived from the summarized texts and those of the original texts, utilizing Spearman's correlation coefficient as the main metric.

**Retrieval:** The main evaluation metric employed in this task is NDCG@10, which assesses the quality of the results based on their relevance and position in the list returned.

# F    COMPLEXITY SCORE CALCULATION

## F.1    SHANNON ENTROPY IN INFORMATION THEORY

The Shannon entropy is a measure from information theory that quantifies the average level of uncertainty or information content inherent in a set of possible outcomes. A higher Shannon entropy means higher uncertainty. To calculate the Shannon entropy $H$ of a text, we follow these steps:

1. Count Tokens: Identify all unique tokens $w_i$ in the text and count their occurrences $n_i$.

2. Calculate Probabilities: Compute the probability of each token $P(w_i)$ by dividing its count by the total number of tokens $N$:

$$P(w_i) = \frac{n_i}{N}, \quad \text{where} \quad N = \sum_{i=1}^{M} n_i$$

Here, $M$ is the total number of unique tokens.

3. Compute Shannon Entropy: Use the probabilities to calculate the entropy, and sum up all unique tokens in the text.

$$H = -\sum_{i=1}^{M} P(w_i) \log_2 P(w_i)$$

### F.2 READABILITY: GUNNING FOG INDEX

The Gunning Fog Index (Gunning, 1952) is a readability metric that estimates the years of formal education needed to understand a text upon first reading, it evaluates the complexity of English prose based on sentence length and the frequency of complex words. To calculate the Gunning Fog Index (GFI), follow these steps:

1. Select a Representative Passage
   Choose at least 1000 words from the text that represents the overall writing style.

2. Calculate the Average Sentence Length (ASL)

$$\text{ASL} = \frac{\text{Total Number of Words}}{\text{Total Number of Sentences}} \tag{2}$$

3. Identify Complex Words

   - **Complex words** are words with **three or more syllables**.
   - Exclude proper nouns, familiar jargon, compound words, and verbs with common suffixes (e.g., "-es", "-ed", "-ing").

4. Calculate the Percentage of Complex Words (PCW)

$$\text{PCW} = \left( \frac{\text{Number of Complex Words}}{\text{Total Number of Words}} \right) \times 100 \tag{3}$$

5. Compute the Gunning Fog Index

$$\text{GFI} = 0.4 \times (\text{ASL} + \text{PCW}) \tag{4}$$

### F.3 MEAN DEPENDENCY DISTANCE

The mean dependency distance (MDD) (Oya, 2011) is introduced as a metric to quantify the syntactic complexity of sentences based on dependency parsing. A higher mean dependency distance indicates longer dependencies and potentially more complex syntactic structures. For each dependency relation between a word (a head) and its dependent in a sentence $d$ is calculated as the absolute difference of their positions in the sentence:

$$d = |\text{Position}_{\text{head}} - \text{Position}_{\text{dependent}}|$$

Here:

- $\text{Position}_{\text{head}}$ is the position index of the head word.
- $\text{Position}_{\text{dependent}}$ is the position index of the dependent word.

The sentence-level MDD is computed by averaging the dependency distances of all its $N$ dependency relations:

$$\text{MDD}_{\text{sentence}} = \frac{1}{N} \sum_{i=1}^{N} d_i = \frac{1}{N} \sum_{i=1}^{N} \left| \text{Position}_{\text{head}_i} - \text{Position}_{\text{dependent}_i} \right|$$

Same with sentence-level, the document-level MDD averages the sentence-level mean dependency distances across all $M$ sentences in the document:

$$\text{MDD}_{\text{document}} = \frac{1}{M} \sum_{j=1}^{M} \text{MDD}_{\text{sentence}_j}$$

In our experiments, we calculate the document-level MDD for a test sample by averaging the MDD of all its text. For example, to compute the MDD for a pair-classification data point, we average the MDD of sentence 1 and sentence 2.

## G  PROMPT FOR CHATGPT ERROR RATE

The detailed example prompt of each task for the ChatGPT Error Rate is listed in Table tables 15 to 21.

Table 15: Prompt for the ChatGPT Error Rate on Semantic Textual Similarity (STS).

Determine whether the following two sentences are similar and answer yes or no.
**Sentence1:** Excluding the impact of merger-related costs, NSTAR Electric s earnings increased $67.4 million in 2013, as compared to 2012, due primarily to lower overall operations and maintenance costs and higher retail electric sales due primarily to colder weather in the first and fourth quarters in 2013.
**Sentence2:** NSTAR Electric's earnings increased in 2014, as compared to 2013, due primarily to lower operations and maintenance costs primarily attributable to lower employee-related costs and higher transmission earnings, partially offset by higher interest expense, higher depreciation expense, higher property tax expense and the after-tax reserve recorded for the 2014 FERC ROE orders as compared to the reserve recorded in 2013 for the FERC ALJ initial decision in the FERC base ROE complaints.

Table 16: Prompt for the ChatGPT Error Rate on Classification.

Classify the sentiment of a given finance text as either positive, negative, or neutral.
**Text:** Glencore shares hit 3-month high after refinancing key credit line

Table 17: Prompt for the ChatGPT Error Rate on Retrieval.

Given a financial question, retrieve user replies that best answer the question. Return the index.
**Query:** What is 'Obligor '?
**Corpus:** {A List of 20 different context}

## H  SUPPLEMENTARY EXPERIMENT: FINETUNE SOTA EMBEDDING USING DOMAIN CORPUS

We fine-tuned e5-mistral-7b-instruct (Wang et al., 2023) using a syntenic finance QA dataset generated through Self-instruct(Wang et al., 2022) with GPT-4o mini (OpenAI, 2024a) and a manually labeled seed finance dataset. The results demonstrate clear domain adaptation effects:

On FinMTEB (Table 22), performance improved from 0.6475 to 0.6735, showing the benefits of finance-specific training. While general MTEB scores (Table 23) slightly decreased from 0.6463 to 0.6320, the model maintained competitive performance on broader tasks.

These results highlight how domain adaptation can enhance specialized task performance while preserving general capabilities.

Table 18: Prompt for the ChatGPT Error Rate on Clustering.

Identify industries from company descriptions.
**Choices:** Banking;Retail;Automotive;Aerospace;Financial services
**Text:** Cobridge Communications was a cable television, high-speed internet, and digital telephone service provider.

Table 19: Prompt for the ChatGPT Error Rate on Reranking.

Given a financial question, retrieve user replies that best answer the question. Return the index.
**Query:** How to tell if you can trust a loan company?
**Corpus:** {A List of 20 different context}

Table 20: Prompt for the ChatGPT Error Rate on Pair-Classification.

Classify the sentiment of a given finance text and determine whether label of two sentences are similar. Answer yes or no.
**Sentence1:** gold falls as dollar strengthens, etf holdings decline
**Sentence2:** Gold futures succumb to profit-booking, global cues

Table 21: Prompt for the ChatGPT Error Rate on Summarization.

Determine whether the following are douments and summary. Answer yes or no.
**Text:** deposits grew $ 167.8 million , or 7 % , to $ 2.504 billion at december 31 , 2020 , compared to $ 2.336 billion at december 31 , 2019. non-interest bearing deposits grew by $ 225.8 million in 2020 , or 20 % , and made up 54 % of total deposits at year-end . cost of deposits remained low at 0.11 % in 2020 , compared to 0.20 % in 2019. net interest income totaled $ 96.7 million and $ 95.7 million in 2020 and 2019 , resp....
**Summary:** cash and cash equivalents : our cash and cash equivalents , which include federal funds sold and short-term investments , were $ 181.5 million at december 31 ....

Table 22: Overall Performance on FinMTEB

| Model | STS | Ret. | Rerank. | Class. | Summ. | PairClass. | Clust. | Avg. |
|---|---|---|---|---|---|---|---|---|
| e5-mistral-7b-instruct | 0.380 | 0.675 | 0.988 | 0.645 | **0.528** | 0.739 | **0.578** | 0.648 |
| + domain adaption | **0.428** | **0.699** | **0.990** | **0.757** | 0.480 | **0.801** | 0.560 | **0.674** |

Table 23: Overall Performance on MTEB

| Model | STS | Ret. | Rerank. | Class. | Summ. | PairClass. | Clust. | Avg. |
|---|---|---|---|---|---|---|---|---|
| e5-mistral-7b-instruct | **0.859** | **0.588** | 0.602 | 0.770 | **0.314** | **0.883** | 0.508 | **0.646** |
| + domain adaption | 0.858 | 0.491 | **0.603** | **0.776** | 0.306 | 0.875 | **0.517** | 0.632 |

