# OpenReview forum: "Do We Need Domain-Specific Embedding Models? An Empirical Investigation"
_ICLR.cc/2025/Conference — Submitted to ICLR 2025_

### Official Review · Reviewer_xpNJ · 2024-11-01

**Soundness:** 3
**Presentation:** 3
**Contribution:** 2
**Rating:** 6
**Confidence:** 3

**Summary:**

This paper investigates the necessity of domain-specific embedding models given the prevalence of general-purpose embedding models. To address this, the authors first developed a financial domain benchmark, FinMTEB. Utilizing this benchmark, they observed a significant decline in performance and, through a series of experiments, ruled out the influence of the selected models and task complexity. This led to the conclusion that general-purpose models are inadequate for domain-specific tasks.

**Strengths:**

- The development of the FinMTEB benchmark provides a valuable tool for evaluating embedding models within the financial domain.
- The experimental design is highly rigorous, systematically eliminating the potential confounding factors related to the models used and the inherent complexity of the datasets. This enhances the credibility and reference value of the experimental conclusions.

**Weaknesses:**

- The conclusion that general-purpose embedding models are not well-suited for specific domain tasks and that specialized models are necessary for particular domains is not particularly surprising.

**Questions:**

None

---

> ### Author Response · Authors · 2024-11-20
>
> Thank you sincerely for taking the time to review our paper and for providing such valuable feedback. We are delighted to hear that you recognize the development of the FinMTEB benchmark and our rigorous experimental design as significant strengths of our work.
>
> While the necessity for domain-specific models might align with reviewer's intuitive expectations, our work still distinguishes itself.
>
> **Why this is surprising:**
>
> 1. current SOTA embedding models are built upon powerful LLMs (e.g., NV-Embed based on Mistral-7B).
>     1. These models are pretrained on extensive, diverse datasets that include domain content.
>     2. Their comprehensive training suggests an inherent capability to understand domain contexts.
> 2. Our testing datasets only contain traditional NLP embedding tasks using financial-related text, rather than specialized financial tasks such as stock prediction. Interestingly, merely changing the domain of texts leads to a performance drop, even when we control for the complexity of the text.
>
> This is why many studies explore domain adaptation for LLMs in generation tasks, such as BloombergGPT (Wu et al., 2023), although it has also been examined in smaller models like FinBert (Yang et al., 2020).
>
> **Why this is useful:**
>
> 1. Answer the question, "Do we still need domain-specific embedding models?" Given that general-purpose embedding models are becoming the backbone of NLP applications, companies like OpenAI and Cohere offer embeddings that potentially serve a wide range of industry applications.
> 2. Training a high-performance domain-specific embedding model costs significant time and money. But there is no strong evidence suggesting that general-purpose SOTA models cannot grasp domain-specific languages or linguistic patterns.
>
> To conclude, our study provides empirical evidence on whether SOTA LLM-based embedding models can effectively handle domain-specific tasks without further specialization. This exploration into embedding models addresses a critical area that has been underinvestigated, thereby contributing valuable insights to the field.
>
> Once again, we sincerely thank you for your valuable feedback and remain available for further discussion if you have any additional questions or require more clarification.

---

> > ### Comment · Reviewer_xpNJ · 2024-11-23
> > **Reply to the authors**
> >
> > Thanks for your reply. I prefer to keep the score unchanged.

---

> ### Author Response · Authors · 2024-11-23
> **Thank You!**
>
> Dear Reviewer xpNJ,
>
> We deeply appreciate the time and effort you have dedicated to reviewing our work.
>
> If you have any specific aspects of our work that would benefit from further clarification, we would be more than happy to provide additional details.
>
>
> Thank you once again, and we wish you all the best!
>
> Authors of paper #9365

---

### Official Review · Reviewer_AfJc · 2024-11-02

**Soundness:** 3
**Presentation:** 3
**Contribution:** 3
**Rating:** 6
**Confidence:** 4

**Summary:**

This paper presents the Finance Massive Text Embedding Benchmark (FinMTEB). As a domain-specific counterpart to the general-purpose MTEB, FinMTEB includes 64 financial datasets curated for the evaluation of embedding models in the financial domain. A comparative analysis of model performance on both MTEB and FinMTEB suggests the need for domain-specific embeddings and benchmarks.

**Strengths:**

- This paper presents a massive financial text embedding benchmark — FinMTEB, which consists of 64 financial datasets. I think it is a helpful resource for the financial NLP research community.
- This paper designs a novel and comprehensive evaluation framework to prove that the performance gap between MTEB and FinMTEB is due to the models’ capabilities rather than inherent dataset complexity.
- The paper is well organized and well written.

**Weaknesses:**

- From my point of view, the title is not good because the main contribution of this paper is the benchmark. Why not refer to the benchmark directly in the title?
- The evidence may be insufficient to support that the general embedding is not good at domain application since only one domain, i.e., finance, is proven in this paper.
- Lack of novelty: According to the previous work [1][2][3], domain fine-tuning can improve domain-specific performance, which is well-accepted evidence. Based on this, the FinMTEB benchmark should be the main contribution of this paper, not the "byproduct" as claimed in the paper.


**Reference:**

[1] Gururangan, S., Marasović, A., Swayamdipta, S., Lo, K., Beltagy, I., Downey, D., & Smith, N. A. (2020). Don't stop pretraining: Adapt language models to domains and tasks. arXiv preprint arXiv:2004.10964.

[2] Huang Y, Wang K, Dutta S, et al. AdaSent: Efficient Domain-Adapted Sentence Embeddings for Few-Shot Classification[J]. arXiv preprint arXiv:2311.00408, 2023.

[3] Schopf T, Schneider D N, Matthes F. Efficient domain adaptation of sentence embeddings using adapters[J]. arXiv preprint arXiv:2307.03104, 2023.

**Questions:**

**Question:**

- From Table 7 to 13, I found that there are part of Chinese datasets in the benchmark. But the part of selected embedding models are English model, e.g. bge-large-en-v1.5 and bge-en-icl. Is this a potential problem?
- How long will it take to benchmark different scale models? It would be better to report the benchmarking time.

**Suggestions:**

- The related work on general-purpose embeddings is insufficient. More pilot and recent LLM sentence embeddings are encouraged to be considered in the related work, such as [1][2][3][4][5].
- According to the ICLR style, a table caption is better placed before the table, and a figure caption is better placed after the figure.

**Reference:**

[1] Li X, Li J. Angle-optimized text embeddings[J]. arXiv preprint arXiv:2309.12871, 2023.

[2] Lee J, Dai Z, Ren X, et al. Gecko: Versatile text embeddings distilled from large language models[J]. arXiv preprint arXiv:2403.20327, 2024.

[3] Li X, Li J. BeLLM: Backward Dependency Enhanced Large Language Model for Sentence Embeddings[C]//Proceedings of the 2024 Conference of the North American Chapter of the Association for Computational Linguistics: Human Language Technologies (Volume 1: Long Papers). 2024: 792-804.

[4] Springer J M, Kotha S, Fried D, et al. Repetition improves language model embeddings[J]. arXiv preprint arXiv:2402.15449, 2024.

[5] Lee C, Roy R, Xu M, et al. NV-Embed: Improved Techniques for Training LLMs as Generalist Embedding Models[J]. arXiv preprint arXiv:2405.17428, 2024.

---

> ### Author Response · Authors · 2024-11-20
>
> Thank you for your insightful and constructive feedback on our paper. We greatly appreciate the time you’ve taken to help us improve our work. Below, we address each of your comments and suggestions.
>
> **1. Title Alignment**
>
> Inspired by the comments, we have revised our title to "Do We Need Domain-Specific Embedding Models? An Empirical Investigation on the Finance Domain and a Domain-MTEB Benchmark," which better captures our contribution.
>
> **2. Lack of novelty**
>
> We understand your concern that the novelty may seem limited given existing studies on domain adaptation. However, our work still distinguishes itself. The previous work [1] [2] [3] focuses on smaller models like DistilRoBERTa; however, the conclusion cannot be applied to the current context. Because current SOTA embedding models are built upon powerful LLMs (e.g., NV-Embed based on Mistral-7B).
>
>     1. These models are pretrained on extensive, diverse datasets that include domain content.
>     2. Their comprehensive training suggests an inherent capability to understand domain contexts.
>
> This is why many studies explore domain adaptation for LLMs in generation tasks, such as BloombergGPT (Wu et al., 2023), although it has also been examined in smaller models like FinBert (Yang et al., 2020).
>
>
> **2. One Domain**
>
> We select the finance domain because financial NLP is a critical area within the research community, with a wealth of established financial NLP datasets. While our current study concentrates on the finance domain, we believe it effectively illustrates the limitations of general-purpose embeddings in specialized contexts. But we plan to include preliminary results from additional domains in future work, which will strengthen the evidence supporting our claims.
>
> ---
>
> **Questions**
>
> **1. Chinese Tasks**
>
> All the experiments in the paper only use the English datasets within FinMTEB (35 English datasets across 7 tasks). We apologize for any confusion and will highlight this point in the paper.
>
> **2. Benchmarking Time Reporting & Expanding Related Work on General-Purpose Embeddings**
>
> We agree that reporting benchmarking time is essential for reproducibility and resource assessment. Therefore, we will add a new section detailing the computational resources utilized and the time required to benchmark models of varying scales.
>
> We also tested the mentioned models and presented the results, including benchmarking times using a batch size of 512. Since Gecko is unavailable, we tested four other methods.
>
> | Model | STS | Retrieval | Reranking | Classification | Summarization | PairClassification | Clustering | Avg | Duration/hr |
> |---------|------|-----------|-----------|----------------|---------------|------------------|------------|------|-------------|
> | Echo Embedding | 0.4380 | 0.6443 | 0.9765 | 0.6525 |0.4722 |0.6261 | 0.5776 | 0.6267 | 12.00 |
> | AnglE-BERT  | 0.3080 | 0.5730 | 0.9650 | 0.6439 | 0.5049 | 0.6891 | 0.5774 | 0.6088 | 8.00 |
> | NV-Embed v2 | 0.3739 | 0.7061 | 0.9822 | 0.6393 | 0.5103 | 0.6043 | 0.6096 | 0.5739 | 5.60 |
> | BeLLM | 0.3919 | 0.0169 | 0.5661 | 0.1168 | 0.3906| 0.5682 | 0.0685 |0.3027 | 11.98 |
>
> The average performance does not surpass the other models reported in the paper.
>
> **3. Compliance with ICLR Style for Captions**
>
> Thanks for pointing out this! We have revised the placement of table and figure captions to adhere to ICLR style guidelines, ensuring that table captions precede the tables and figure captions follow the figures.
>
> We appreciate your thorough review and remain available for further discussion if you have any additional questions or require more clarification.

---

> > ### Comment · Reviewer_AfJc · 2024-11-24
> > **Reply to the authors**
> >
> > Thank you very much for providing the additional experimental results. Your responses have addressed my concerns, but I prefer to maintain my original score. I suggest that the author consider revising the title and including these experiments in the final version.

---

> > > ### Author Response · Authors · 2024-11-24
> > > **Thank You!**
> > >
> > > Dear Reviewer AfJc,
> > >
> > > We deeply appreciate the time and effort you have dedicated to reviewing our work, as well as your prompt response.
> > >
> > > We have revised the title and will consider including these experiments in the final version. If you have any specific aspects of our work that would benefit from further clarification, we would be more than willing to provide additional details.
> > >
> > > Thank you once again, and we wish you all the best!
> > >
> > > Authors of paper #9365

---

### Official Review · Reviewer_WW39 · 2024-11-02

**Soundness:** 2
**Presentation:** 2
**Contribution:** 2
**Rating:** 3
**Confidence:** 4

**Summary:**

This paper investigates the performance of text embedding models on two distinct benchmarks: a general-domain benchmark and a finance-specific benchmark, which is a newly collected dataset. The findings highlight a performance gap and lack of correlation between the two, suggesting the need for domain-specific benchmarks and embedding models better to capture specialized linguistic patterns in domains like finance.

**Strengths:**

This paper explores a critical question―whether domain-specific embedding models are necessary in the era of general-purpose, large-scale language models.
This paper presents the Finance Massive Text Embedding Benchmark (FinMTEB), a collection of existing or newly constructed evaluation datasets.
The dataset may be useful for emphasizing the finance-specific abilities in benchmarking because the performances on the (general-domain) MTEB and FinMTEB seem less correlated.
This paper evaluates dataset complexity in addition to model performances.

**Weaknesses:**

The paper’s title question, "Do We Need Domain-Specific Embedding Models?" is not convincingly addressed. To conclude that domain-specific models are necessary, it would be important to demonstrate that (1) a domain-specific embedding model can outperform a general-purpose model within that domain and (2) the observed difference is specifically due to domain specialization. However, the paper does not provide such evidence. Therefore, the research question (and title) would be better phrased as "Do We Need Domain-Specific Embedding Benchmarks?" or "Are Current SoTA Embedding Models Sufficient for Domain-Specific Tasks?"

The analysis lacks sufficient logical rigor and is sometimes misleading. For example, describing the performance difference between FinMTEB and MTEB as a "significant performance drop" is misleading, as MTEB and FinMTEB are not directly comparable. These benchmarks differ in various aspects, making it speculative to attribute the performance difference primarily to financial domain knowledge. A clearer attribution of the observed effects is necessary. Even with additional analyses, it remains unclear whether any convincing conclusions can be drawn about the models solely based on this performance comparison.

**Questions:**

The paper claims that inter-dataset similarity scores below 0.6 indicate high semantic diversity. Could the authors clarify if this threshold reliably indicates diversity?

How were datasets divided into low, medium, and high complexity? Were the same thresholds used across datasets, or were they divided within each dataset independently? Further clarification here would enhance the validity of this categorization.

ChatGPT Error Rate as a Complexity Metric: Could the authors provide more details on how ChatGPT's error rate was used to categorize dataset complexity? Specifically, how was this metric interpreted to distinguish three complexity levels? An ChatGPT's judge (correct or wrong) might be able to be aligned with binary complexity in my understanding.

FinMTEB’s Dataset Source: In Tables 7 to 13, is the lack of citations for certain datasets indicative of new dataset construction? Clarifying this would help in assessing the novelty of FinMTEB.

In Section 5, the interpretation of p-values could be misleading. A large p-value indicates a lack of evidence for a relationship, not proof of no relationship. Rephrasing to clarify this distinction would avoid statistical misinterpretation.

The abstract mentions that the analysis provides "compelling evidence" of difficulties in capturing domain-specific patterns. Could the authors clarify where this evidence is presented in the analysis and summarize them, as it was not immediately apparent?

---

> ### Author Response · Authors · 2024-11-20
>
> Thank you for your thoughtful and constructive feedback on our paper. We sincerely appreciate the time you've dedicated to providing insights that help us clarify and enhance our work. Below, we address each of your comments and questions.
>
> **For Weaknesses:**
>
> **1. Title Clarity**
> Inspired by the comments, we have revised our title to "Do We Need Domain-Specific Embedding Models? An Empirical Investigation on the Finance Domain and a Domain-MTEB Benchmark," which better captures our contribution.
>
> **2. Analysis Logic**
> Your concern that "MTEB and FinMTEB are not directly comparable" aligns with our motivation for further analysis. This is precisely why we controlled for text complexity before comparing performance statistically, given that we do not have a domain-specific embedding model.
>
> The controlled experiment makes the comparison valid because:
> 1. We focus on four different aspects of text complexity across 7 SOTA models, 7 tasks, and 35 datasets.
> 2. We use different regression analysis to statistically show a significant performance gap between embedding models on MTEB and FinMTEB, even when controlling for the same level of dataset complexity.
> 3. We also find that general embedding models struggle with the highest complexity domain datasets, while they can handle the highest level of general text. This could be an interesting point for improving the domain-specific embedding training process.
>
> ---
> **For Questions:**
>
> **1. Semantic Diversity**
>
> **Mean Cosine Similarity of 0.4:** We apologize for any confusion caused by our previous statements. The threshold of 0.6 mentioned in the paper is not the mean value of inter-dataset cosine similarities. It represents the maximum observed cosine similarity value.
>
> For the FinMTEB benchmark, the mean cosine similarity across all datasets is 0.3746. This value was computed by embedding 1000 samples from each dataset and then calculating the pairwise cosine similarities between these embeddings. This low mean cosine similarity suggests that, on average, the datasets in our benchmark do not have large semantic overlap. We appreciate your feedback and will revise the statement in the paper to avoid any misleading information.
>
> **2. Division of Datasets into Low, Medium, and High Complexity**
>
> We divided the data samples into three equal-sized groups (tertiles) based on the range of complexity scores shared by both benchmarks. For example, the readability score range of FinMTEB is [5:15), and for MTEB it is [1:12.5). Therefore, we divided the readability scores into the following levels: [5:7.5), [7.5:10), and [10:12.5).
>
> **3. ChatGPT Error Rate as a Complexity Metric**
>
> We appreciate the opportunity to elaborate:
>
> - **Methodology:** We reformatted dataset examples into instruction-question-answer (IQA) pairs, as shown in Appendix F. Here, ChatGPT functions as an LLM-based retriever or classifier, utilizing question-answering prompts to locate relevant information.
>
> This metric primarily captures the complexity in query-document relationships, serving as a complementary tool to other traditional metrics. Traditional metrics often struggle to capture the nuanced relationship between queries and documents, making the ChatGPT error rate a valuable addition.
>
> **4. FinMTEB's Dataset Sources**
>
> Most datasets without explicit citations in Tables 7-13 were cited in earlier sections. However, two types of datasets were newly constructed for this paper:
>
> 1. For the finance terminology retrieval task: We constructed TheGoldman dataset from the Goldman Sachs Financial Dictionary.
> 1. For the clustering task: We constructed FinanceArxiv-s2s, FinanceArxiv-p2p by titles and abstracts from the arXiv.
>
> We apologize for any confusion regarding dataset citations and will clarify this in the revision.
>
> **5. Interpretation of p-values in Section 5**
>
> You are absolutely correct in your observation. We acknowledge the misinterpretation of p-values in our initial submission. We have revised our interpretations to accurately reflect that a large p-value indicates insufficient evidence to reject the null hypothesis, ensuring the statistical analyses are correctly presented.
>
> From "no relationship" to "a lack of evidence for a relationship."
>
> **6. Evidence of Challenges in Capturing Domain-Specific Patterns**
>
> We understand the concerns, and to clarify the evidence:
>
> - **Performance Gap with Controlled Variables:** SOTA embedding models consistently underperform on FinMTEB compared to MTEB across various tasks, even after controlling for dataset complexity.
>
> - **Complexity-Dependent Performance Degradation:** These models show greater performance drops on high-complexity financial tasks, highlighting difficulties in capturing complex, domain-specific language and semantics.
>
> Once again, we sincerely thank you for your valuable feedback and remain available for further discussion if you have any additional questions or require more clarification.

---

> > ### Comment · Reviewer_WW39 · 2024-11-22
> >
> > Weakness 1
> > Thank you. However, my concern might not have been fully conveyed. The main issue is not simply that the study is limited to a single domain but rather that it focuses on datasets, not models. The current analysis does not address the central question posed by the title, "Do We Need Domain-Specific Embedding Models?" To sufficiently answer this, it would be important to demonstrate two key points: (1) that a domain-specific embedding model outperforms a general-purpose model within the same domain, and (2) that this performance difference is specifically due to domain specialization. These points are not addressed in the current paper.
> >
> > Weakness 2
> > I appreciate your efforts to address complexity metrics and comparability between MTEB and FinMTEB. However, the metrics provided do not clearly establish that these datasets are comparable. Without robust evidence that the datasets are comparable, it is difficult to conclude that the observed performance differences are significant or relevant to the paper's central question. Moreover, even if a general-domain model performs worse on a domain-specific dataset, this alone does not answer whether domain-specific embedding models are necessary.
> > Possibly, the definition of domain-specific embedding models could be different. Defining "domain-specific embedding models" could be a good clarification.
> >
> > ---
> >
> > Q1
> > Thank you for elaborating on the cosine similarity threshold. However, it remains unclear whether a mean similarity of 0.3746 reliably indicates diversity. For instance, how would we interpret values like 0.4 or 0.5? Without clear distinctions or benchmarks to compare against, determining the reliability of an absolute value's meaning is difficult. A comparison with other datasets or additional context might help clarify its validity.
> >
> > Q3
> > I may have misunderstood your approach. In my understanding, the ChatGPT test produces a binary output ("Correct" or "Error") for each question (e.g., classification tasks). If this is the case, how is the error rate of a question defined continuously? If the classification is binary, it seems challenging to divide the results into tertiles. I would appreciate more clarification on this point.
> >
> > Q6
> > Thank you for your response. However, it mainly demonstrates that general-purpose models perform worse on FinMTEB than on MTEB. This macro-level comparison does not provide "compelling evidence" about models' ability (or inability) to "capture domain-specific patterns." Could the authors define what "Domain-Specific Patterns" entail, provide concrete examples, and analyze models' behavior concerning these patterns? Such an analysis would strengthen the paper's scientific contribution and clarity.

---

> ### Author Response · Authors · 2024-11-22
>
> Thanks for your prompt feedback! To better address your concerns, we reorganized our response to your questions as follows:
>
> **1. Q6 & Weakness 2**
>
> First, in our work, **domain-specific patterns** represent two distinct aspects:
>
> - **Terminology:** This includes domain-specific terms and expressions.
> - **Domain Document Format:** This pertains to the structure and word usage patterns typical of financial documents. For example, in the finance domain, the text in annual reports often follows specific formats.
>
> The FinMTEB datasets contains these patterns, for example:
>
> - TheGoldman: This is a finance terminology retrieval dataset.
> - FinSTS: This is an STS (Semantic Textual Similarity) dataset that compares annual reports from the same company over different years.
> - FinSent: This sentiment classification dataset uses financial news to determine sentiment.
> - WikiCompany2IndustryClustering: Clustering the company description by industry. (Collected from Wikipedia)
>
> These datasets do not require extensive financial knowledge such as stock prediction or asset recommendation. They are categorized as domain-specific tasks primarily because they involve financial texts. But the objectives of these tasks are still traditional NLP embedding tasks.
>
> Thus, **a domain-specific embedding model** is one adapted using a target domain corpus.
>
> For example, _if a medical company wants to develop a RAG system, is it necessary for them to train an embedding model using medical documents, such as Medrxiv, given the recent progress in LLM-based embeddings?_
>
> The pre-training process of LLMs may have already included such documents, and training a new model requires money and time.
>
>
> **2. Weakness 1**
>
> To better address your concern, we fine-tuned e5-mistral-7b-instruct (Wang et al., 2023) using a syntenic finance retrieval dataset (no overlap with FinMTEB).
>
> - **Overall Performance:**
>
> | Model                                | STS   | Ret.  | Rerank. | Class. | Summ. | PairClass. | Clust. | **FinMTEB Avg.** |
> |--------------------------------------|-------|-------|--------|--------|-------|-----------|--------|------------------|
> | e5-mistral-7b-instruct (finance) | **0.4281** | **0.6989** | **0.9896** | **0.7565** | 0.4797 | **0.8014** | 0.5600 | **0.6735**        |
> | e5-mistral-7b-instruct               | 0.3800 | 0.6749 | 0.9875 | 0.6449 | **0.5275** | 0.7394    | **0.5783** | 0.6475           |
>
>
> | Model                                | STS   | Ret.  | Rerank. | Class. | Summ. | PairClass. | Clust. | **MTEB Avg.** |
> |--------------------------------------|-------|-------|--------|--------|-------|-----------|--------|---------------|
> | e5-mistral-7b-instruct (finance) | 0.8582 | 0.4905 | **0.6025** | **0.7758** | 0.3060 | 0.8747    | **0.5166** | 0.6320        |
> | e5-mistral-7b-instruct               | **0.8594** | **0.5875** | 0.6021 | 0.7700 | **0.3140** | **0.8834**    | 0.5077 | **0.6463**        |
>
> - **Continue with the examples in Q6:**
>
> | Model                                | TheGoldmanEn | FinSTS | FinSent | WikiCompany2Industry |
> |--------------------------------------|-------|-------|--------|--------|
> | e5-mistral-7b-instruct (finance) |0.6887 |0.2988 | 0.7178 | 0.7636
> | e5-mistral-7b-instruct               |0.5780 |0.2677 | 0.5591 | 0.7360
>
> It is clear that using domain-specific documents to fine-tune the general model increases performance on the domain datasets, but slightly decreases performance on the general datasets. We will provide these experiments in detail in the final revised paper to supplement our argument.
>
> ---
> **3. Q1**
>
> You are right that mean cosine similarity may not fully represent semantic diversity. In section 3.2, we calculate cosine similarity to align with MTEB (Muennighoff et al., 2022), which also presented cosine similarity in a heatmap. The FinMTEB heatmap is provided in Appendix A. From the heatmap, we observe that datasets with the same task objective may show high cosine similarity, but the overall similarity is not necessarily very high.
>
> We will revise the statement in Section 3.2 for clarity, deleting the statement about 'high semantic diversity'. Since our benchmark comprises 64 datasets that already represent the overall diversity, we will showcase our diversity from this perspective.
>
> **4. Q3**
>
> The role of GPT isn't the judge. It is an LLM embedder. For example, for a retrieval task, we will provide the GPT with the following prompt:
> ```
> Given a financial question, retrieve user replies that best answer the question. Return the index.
> Query: What is ’Obligor ’?
> Corpus: {A List of 20 different context}
> ```
> We will judge the answer returned by GPT for all tasks. The error rate is calculated at the task level in a bootstrap way, as illustrated in Section 3.3.1. We will emphasize this in the revised paper.
>
> It's inspiring to discuss with you! We remain available for further discussion if you require more clarification.

---

> > ### Comment · Reviewer_WW39 · 2024-11-23
> >
> > Thank you for the further reply! I truly appreciate the additional definitions, as they could be very helpful for clarifying discussions. Additionally, the fine-tuning experiment is a valuable addition and provides some meaningful insights.
> >
> > However, I feel the contribution might still differ slightly from the resolution of my original concern. If the definition of "Domain-Specific Embedding Models" is "one adapted using a target domain corpus," the contribution may remain somewhat limited. The additional experiment you presented could effectively address the question, "Can an Additional (Domain-Specific) Training Dataset be Helpful (for the Current SoTA Embedding Models)?" However, in the machine learning and NLP community, the answer to this question might already be considered apparent. If this question is not as obvious as it seems, providing further explanations in the paper about why it is not obvious would greatly strengthen the contribution.
> >
> > As an intriguing contribution, this study might be expected (I had expected before reading the manuscript) to explore a more nuanced question: "Do current SoTA large embedding models face the no-free-lunch theorem, where improving performance on domain-specific tasks comes at the cost of performance on general-purpose tasks?" If such tradeoffs are demonstrated, they could potentially be attributed to model capacity limitations or conflicting requirements across domains, adding deeper insights into the challenges of domain-specific adaptation in the large-scale model era. This perspective could further elevate the significance of this work.

---

> > > ### Author Response · Authors · 2024-11-23
> > >
> > > Thank you very much for your clarification and patient explanation once again!
> > >
> > > **Q1. RQ**
> > >
> > > The question, "Can Domain-Specific Additional Training Data Improve SoTA Embedding Models?" remains an open and non-trivial research question for several reasons:
> > >
> > > * **Necessity of Research on the 'Importance of Domain Adaptation':**
> > >
> > >   - Earlier studies have attempted to demonstrate the importance of domain adaptation in smaller models such as RoBERTa [1].
> > >   - Recent studies also try to show meaningful improvements in various LLM generation tasks [2]
> > >
> > > * **Limited Research on Domain Adaptation for LLM Embeddings:**
> > >
> > >   - Current SoTA embedding models are primarily built from general-purpose LLMs that have been trained on vast text corpora covering nearly every domain
> > >   - No existing work has attempted to prove the importance of domain adaptation in LLM embedding tasks. If the answer to this question were obvious, there wouldn't be so much research exploring domain adaptation in other tasks.
> > >   - We believe that our main experiments, along with the additional experiments you suggested, will provide strong evidence for this question.
> > >   - FinMTEB: The proposed benchmark will also be a valuable testbed for further exploration.
> > >
> > > We have already demonstrated these reasons in the introduction and related work sections. However, our presentation may not be explicit enough, so we will refine it for better clarity.
> > >
> > > **Q2. For the nuanced question**
> > >
> > > The research question you proposed is indeed interesting and we will keep explore it in the future direction.
> > >
> > > **However, it is essential first to establish whether domain adaptation can improve the current SoTA embedding models.** This foundational step is crucial to paving the way for examining more nuanced questions. By confirming the benefits of domain adaptation, we can then systematically investigate the model capacity limitations and conflicting requirements across domains.
> > >
> > > For example, AdaptLLM's [3] research follows a logical progression:
> > >   - First, they acknowledge the established importance of domain adaptation from previous work
> > >   - Then, they focus on innovating and improving domain adaptation methods for generation tasks
> > >
> > > Thus, our work paves the way for further exploration in this field. Still, the idea you proposed is very interesting!
> > >
> > >
> > > Lastly, thank you for your careful review. We are happy to address any additional questions. We would appreciate if you could consider updating the overall assessment score based on our responses, if possible. Thank you very much!
> > >
> > >
> > > ---
> > > [1] Gururangan, S., Marasović, A., Swayamdipta, S., Lo, K., Beltagy, I., Downey, D., & Smith, N. A. (2020). "Don't stop pretraining: Adapt language models to domains and tasks." arXiv preprint arXiv:2004.10964.
> > >
> > > [2] Ling, Chen, et al. "Domain Specialization As the Key to Make Large Language Models Disruptive: A Comprehensive Survey." arXiv preprint arXiv:2305.18703.
> > >
> > > [3] Cheng, Daixuan, et al. "Adapting Large Language Models to Domains via Reading Comprehension." arXiv preprint arXiv:2309.09530.

---

> > > > ### Author Response · Authors · 2024-11-27
> > > > **Any further questions?**
> > > >
> > > > Dear Reviewer WW39,
> > > >
> > > > We appreciate your constructive feedback. As noted in our engagement, we believe this paper constitutes a valuable resource for benchmarking domain embedding models across 64 different datasets, as well as paving the way for further, more nuanced questions, such as the one you proposed. We will definitely delve further in this direction.
> > > >
> > > > We hope we have clarified this point and would appreciate it if you could reconsider your rating. Should you have any further questions, we are very happy to address them!
> > > >
> > > > Best, Authors

---

> ### Author Response · Authors · 2024-12-03
>
> Dear Reviewer WW39,
>
> We sincerely appreciate your insightful comments on FinMTEB, and we would like to make one last effort to address the misunderstandings raised regarding our paper. We understand that our rebuttal may require some of your valuable time, so we have provided a brief summary of the key points addressed in our response:
>
>
> **Main Contributions**:
>
> 1. **FinMTEB Benchmark**: We introduced FinMTEB, a finance embedding benchmark with 64 datasets and 7 tasks. We appreciate that reviewers **AfJc** and **xpNJ** have noted it as a valuable and helpful resource. This benchmark aims to advance research in domain-specific embeddings.
>
> 2. **Exploring the Necessity of Domain-Specific Embeddings**: To our knowledge, our study is the first to explore whether domain-specific embeddings are still necessary, given the rise of general-purpose LLM-based embedding models. We used a very comprehensive benchmark and conducted an empirical analysis to address this question.
>
> **Necessity of this Research Question**:
>
> 1. **Limited Research on LLM Embeddings**: While domain adaptation in smaller models like RoBERTa has shown improvements, there's limited research on domain adaptation for LLM embeddings.
>
> 2. **Strong Generalization Ability of LLM Embeddings**: Modern LLM embeddings, built on diverse text corpora, have strong generalization ability. They can be further enhanced through in-context learning (ICL) (as noted by the reviewer **DoPN**). Their strong generalization ability allows them to handle a variety of embedding tasks, which underscores the importance of our research question.
>
> 3. **Practical Implication**: Training a high-performance domain-specific embedding model requires significant time and financial resources. However, there is no strong evidence suggesting that general-purpose state-of-the-art (SOTA) models cannot effectively grasp domain-specific languages or linguistic patterns.
>
> We hope these updates address your concerns and wish you all the best.
>
> Thank you!
>
> Kind regards,
> The Authors

---

### Official Review · Reviewer_DoPN · 2024-11-07

**Soundness:** 3
**Presentation:** 3
**Contribution:** 2
**Rating:** 6
**Confidence:** 3

**Summary:**

This paper introduces FinMTEB, a massive text embedding benchmark specifically designed for the financial domain. Experimental results show that state-of-the-art models that perform well on MTEB exhibit a significant performance drop when evaluated on FinMTEB. The authors further quantify the complexity of both FinMTEB and MTEB and conduct control experiments. They find that, at the same complexity level, model performance on FinMTEB is much lower than on MTEB, which they argue provides evidence that the challenges arise from domain-specific linguistic and semantic patterns rather than dataset complexity. Additionally, the authors observe that performance on MTEB does not correlate well with performance on FinMTEB, highlighting the need for domain-specific embedding benchmarks.

**Strengths:**

1. **Valuable problem**: This paper focuses on a practical and important problem: creating high-quality text embeddings for the financial domain.

2. **Well-written paper**: The paper is clearly written and easy to follow.

3. **Comprehensive model evaluation**: The authors evaluate a diverse range of embedding models, providing a thorough overview of the current embedding model landscape. Additionally, the experimental results demonstrate the challenges posed by FinMTEB, highlighting the need for developing better financial embedding models.

**Weaknesses:**

1. **Should current general-purpose text embedding models perform well on financial embedding tasks?**: I have concerns about whether current SOTA text embedding models should be expected to perform effectively on financial embedding tasks. For example, the NV-Embed [1] model is trained on only a single finance-related dataset, FiQA. Given the limited financial data used in training, these embeddings may not perform well for finance-specific tasks.

2. **Unfair dataset language composition**: Most SOTA general-purpose text embeddings on the MTEB benchmark, including the models evaluated in this paper, are predominantly trained on English datasets. However, a substantial portion of FinMTEB is in Chinese, creating an imbalanced setting that may not be fair for evaluating these models.

3. **Dataset complexity measurement**: I have concerns about the reliability of the authors’ approach to quantifying dataset complexity. A) Complexity is measured independently for queries and documents, but retrieval is fundamentally a pairwise task, where the relationship between query and document can significantly affect complexity. For example, the BRIGHT [2] dataset requires reasoning to map queries to documents, which current models struggle with. B) I question the validity of using ChatGPT’s error rate as a complexity measure. If ChatGPT cannot answer a question correctly, is it due to the complexity of the question or the model’s lack of question-specific knowledge?

4. **Concerns about the argument of linguistic pattern**: I have concerns about the authors' claim that the higher token count in FinMTEB compared to MTEB indicates significant linguistic differences between the two benchmarks.

5. **High semantic diversity**: The inter-dataset semantic similarity is calculated using only 100 samples from each dataset. I question whether this sample size is sufficient to accurately represent the overall dataset’s characteristics.

6. **Unclear presentation of results**: This is a minor point, but only the average score across all FinMTEB tasks is shown. Without detailed results for each task type (e.g., clustering, retrieval), the performance of models on FinMTEB is not fully clear.

[1] Chankyu Lee, Rajarshi Roy, Mengyao Xu, Jonathan Raiman, Mohammad Shoeybi, Bryan Catanzaro, Wei Ping. NV-Embed: Improved Techniques for Training LLMs as Generalist Embedding Models. arXiv preprint arXiv:2405.17428, 2024.

[2] Hongjin Su, Howard Yen, Mengzhou Xia, Weijia Shi, Niklas Muennighoff, Han-yu Wang, Haisu Liu, Quan Shi, Zachary S. Siegel, Michael Tang, Ruoxi Sun, Jinsung Yoon, Sercan O. Arik, Danqi Chen, Tao Yu. BRIGHT: A Realistic and Challenging Benchmark for Reasoning-Intensive Retrieval. arXiv preprint arXiv:2407.12883, 2024.

**Questions:**

Please see my comments in the Weakness section.

---

> ### Author Response · Authors · 2024-11-20
>
> Thank you for your thoughtful and constructive feedback! We appreciate the recognition of the value of our work in addressing the important challenge of financial domain embeddings. Below, we address each of your concerns:
>
> **1. Should Current General-Purpose Text Embedding Models Be Expected to Perform Well on Financial Embedding Tasks?**
>
> Before conducting this research, we hypothesized that state-of-the-art (SOTA) general embedding models would perform well on financial texts. Because current SOTA embedding models are built upon powerful LLMs (e.g., NV-Embed based on Mistral-7B). Thus:
>
>    1. These models are pre-trained on extensive, diverse datasets that include domain content
>    2. Their comprehensive training suggests an inherent capability to understand domain contexts
>
> Our testing datasets only contain traditional NLP embedding tasks using financial-related text rather than specialized financial tasks such as stock prediction. Interestingly, merely changing the domain leads to a performance drop, even when we control the complexity of the text.
>
> Besides, to our knowledge, many works prove the necessity for domain adaptation in generation tasks, but no research has systematically kept up with the recent progress in the field of embeddings.
>
> **2. Evaluation Language Problem**
> All the experiments in the paper only use the English datasets within FinMTEB (35 English datasets across 7 tasks). We apologize for any confusion and will highlight this point in the paper.
>
> **3. Dataset Complexity Measurement**
>
> A) Pairwise Complexity in Retrieval Tasks:
>
> You are correct that retrieval tasks involve the interplay between queries and documents, and complexity can arise from their relationships, that's why we use ChatGPT's error rate as one of the metrics. ChatGPT acts as an LLM-based retriever or classifier, functioning similarly to embedding models but utilizing QA prompts to find the relevant document. This process also involves reasoning.
>
> B) ChatGPT Error Rate as a Complexity Measure:
>
> We acknowledge that ChatGPT’s high error rate may result from its knowledge limitations. However, by categorizing all scores into three difficulty levels using a common scoring range for both general and domain-specific datasets, we ensure that the model’s knowledge limitations are represented consistently across both. Therefore, comparing error rates reliably reflects the relative complexity of the questions, as the model’s limitations equally impact both sets.
>
> While these are valid points, our approach remains robust because:
>
> - **Multiple Metrics:** We don't rely solely on ChatGPT's error rate. We also use semantic entropy, readability, and dependency distance to capture various aspects of complexity.
> - **Task Versatility:** Our metrics apply to a range of 7 different tasks; only the retrieval and reranking depend on pairwise complexity.
>
> **4. Concerns about the Argument of Linguistic Pattern**
>
> **Source of Argument:** We acknowledge your concern regarding the complexity of the linguistic pattern argument. However, this argument is based on multiple factors, including sentence length, token length (average number of characters per token), syllables per token, and mean dependency distance. All metrics statistically show that FinMTEB is more complex than MTEB, as illustrated in Table 1.
>
> **Token Length:** Besides, the higher average number of characters per token means longer words in FinMTEB. Longer words are generally more challenging to process for embedding models.
>
> **5. Semantic Diversity**
> Thank you for pointing this out. While our initial sample size of 100 was chosen to align with MTEB (Muennighoff et al., 2022), we've expanded our analysis to 1,000 samples per dataset (up from 100). The results remain consistent:
> - Mean: 0.3743
> - Std: 0.2635
>
> We also updated the heatmap in Appendix A of the paper using data from 1000 samples.

---

> > ### Author Response · Authors · 2024-11-20
> >
> > **6. Full Result Presentation**
> >
> > We appreciate this feedback and agree that providing detailed results for each task type would offer more comprehensive insights into the models' performance. In the revised paper, we will include detailed performance breakdowns for each task category, such as clustering, retrieval, and classification. We present the detailed table here for reference.
> >
> > | Model                  | STS   | Ret.  | Rerank | Class. | Summ. | PairClass | Clust. | Avg.  |
> > |------------------------|-------|-------|--------|--------|-------|-----------|--------|-------|
> > | bge-large-en-v1.5      | 0.34  | 0.65  | 0.98   | 0.64   | 0.20  | 0.74      | 0.57   | 0.59  |
> > | gte-Qwen1.5-7B-instruct| 0.38  | 0.67  | 0.99   | 0.64   | 0.24  | 0.70      | 0.59   | 0.60  |
> > | text-embedding-3-small | 0.33  | 0.66  | 0.98   | 0.64   | 0.51  | 0.60      | 0.58   | 0.61  |
> > | e5-mistral-7b-instruct | 0.38  | 0.67  | 0.99   | 0.64   | 0.53  | 0.74      | 0.58   | 0.65  |
> > | all-MiniLM-L12-v2      | 0.31  | 0.57  | 0.97   | 0.60   | 0.09  | 0.72      | 0.55   | 0.54  |
> > | instructor-base        | 0.37  | 0.58  | 0.97   | 0.62   | 0.15  | 0.61      | 0.53   | 0.55  |
> > | bge-en-icl             | 0.32  | 0.68  | 0.99   | 0.66   | 0.52  | 0.67      | 0.57   | 0.63  |
> >
> > We appreciate your thorough review and remain available for further discussion if you have any additional questions or require more clarification!

---

> > > ### Author Response · Authors · 2024-11-25
> > >
> > > Dear Reviewer, Thank you for handling our manuscript and providing valuable feedback. We hope that our responses have sufficiently addressed the concerns you raised. We welcome more discussion if you have more questions and suggestions. As the discussion deadline is approaching, we would be very grateful if you could take a moment to review our reply.

---

> > > > ### Comment · Reviewer_DoPN · 2024-11-30
> > > > **Response to Authors**
> > > >
> > > > Thank you for your response, and I sincerely apologize for the delayed reply.
> > > >
> > > > (1) I am afraid I cannot fully agree with this due to the following reasons:
> > > >
> > > > (i) Previous works, such as BEIR [1], have shown that text embedding models perform well on in-domain data but generalize poorly to out-of-domain data, sometimes even underperforming BM25.
> > > >
> > > > (ii) Text embedding is a distinct task from language modeling (e.g., text generation). Is there any evidence that pre-trained models can perform well without in-domain data? A recent study [2] introducing more in-domain MTEB data (e.g., classification data) has been shown to significantly improve MTEB performance. This suggests that while LLMs are good at understanding and have seen many different domains, their embedding ability may still lack generalizability.
> > > >
> > > > (2) In Tables 8, 9, 10, 11, 12, 13, and 14 of Appendix D, many of the datasets used in this work are in Chinese. Could you clarify this?
> > > >
> > > > (3) Thank you for your explanation. My concern has been addressed.
> > > >
> > > > (4) Thank you for your explanation. My concern has been addressed.
> > > >
> > > > (5) Thank you for conducting the new experiments. My concern has been addressed.
> > > >
> > > > (6) Thank you for providing the results. My concern has been addressed.
> > > >
> > > > [1] Nandan Thakur, Nils Reimers, Andreas Rücklé, Abhishek Srivastava, Iryna Gurevych. BEIR: A Heterogenous Benchmark for Zero-shot Evaluation of Information Retrieval Models. NeurIPS 2021 Dataset and Benchmark Track. 2021
> > > >
> > > > [2] Chaofan Li, MingHao Qin, Shitao Xiao, Jianlyu Chen, Kun Luo, Yingxia Shao, Defu Lian, Zheng Liu. Making Text Embedders Few-Shot Learners. Arxiv. 2024.

---

> > > > > ### Author Response · Authors · 2024-11-30
> > > > > **[ We are open to discuss! ] Response to DoPN (Part 1/2 )**
> > > > >
> > > > > We appreciate the reviewer's further feedback and would like to address it in the following points:
> > > > >
> > > > > ---
> > > > >
> > > > > **1. Clarification on Domain:**
> > > > >
> > > > > First of all, we want to clarify that the 'domain' in our paper refers to expert domain text rather than general text from different sources. The term 'out-of-domain' refers to a domain shift from a general corpus to a domain-specific corpus, such as finance.
> > > > >
> > > > > For example, in FinMTEB, we have some clustering tasks sourced from financial news such as FinNL. In MTEB, they also have clustering tasks sourced from general news such as TwentyNewsgroupsClustering.
> > > > >
> > > > >
> > > > > **2. Domain Definition in Previous Works:**
> > > > >
> > > > > * **BEIR:**
> > > > >
> > > > > The domain in BEIR is more aligned with the text source, different from our focus on expertise domains. Here is the domain used in the BEIR paper:
> > > > >
> > > > > Wikipedia, Finance, Twitter, News, StackEx., Misc., Scientific (Only Retrieval Tasks)
> > > > >
> > > > > * **Domain Definition in "Making Text Embedders Few-Shot Learners":**
> > > > >
> > > > > The same with the BEIR.
> > > > >
> > > > > wiki, web, news, healthcare, law, finance, arxiv, msmarco (Only Retrieval Tasks)
> > > > >
> > > > > **3. Comparision With Both Previous Works:**
> > > > >
> > > > > * For both works, the domain definitions are not completely consistent with our work. For example, they classify Wikipedia and the web as two separate domains. They are mainly classified based on different text sources instead of expert knowledge. Our work contains many datasets from various sources, but the knowledge domain remains consistent.
> > > > >
> > > > > * They only evaluated the performance on the retrieval task and did not consider other embedding tasks.

---

> ### Author Response · Authors · 2024-11-30
> **[ We are open to discuss! ] Response to DoPN (Part 2/2 )**
>
> >  1. Should Current General-Purpose Text Embedding Models Be Expected to Perform Well on Financial Embedding Tasks?
>
> (i) Previous works, such as BEIR [1]:
> * **The BEIR only focuses on the retrieval task**, and the definition of domains may be different.
> * **Different Model Scales**: The BEIR highlighted issues with dense retrieval models (e.g., ANCE, TAS-B) for out-of-distribution data but did not address the performance of LLM embedding models in expertise domains.
>
> Recent studies show that domain adaptation, such as training on finance corpora, improves LLM generation tasks [1]. However, no work has focused on domain adaptation in LLM embeddings.
>
> ---
> (ii) There are two questions.
>
> ```
> Is there any evidence that pre-trained models can perform well without in-domain data?
> ```
>
> If we understand correctly, 'pre-trained models' refers to embedding models trained on data that is unrelated to the evaluation corpus. Using only pre-trained models without adaptation to specific embedding tasks will likely result in poor performance.
>
> * **Generality Performance in General Domain**
>
> For general tasks, pre-trained models can indeed perform well. For example, the e5-mistral-7b-instruct[2] model, a well-known LLM-based embedding, has demonstrated superior performance in various tasks such as STS and classification within the MTEB benchmark. The training data for this model included some retrieval tasks from a general corpus, yet it managed to achieve high performance across multiple general-domain tasks.
>
> The training dataset of e5-mistral-7b-instruct[2]: ELI5, HotpotQA, FEVER, MIRACL, MrTyDi, NQ, SQuAD, TriviaQA, NLI, MS-MARCO, Quora Duplicates, DuReader, T2Ranking
>
> * **Generality Performance in Special Domain**
>
> For domain-specific tasks such as finance, the answer is not obvious. To our knowledge, prior to this work, there were no state-of-the-art finance embedding models or extensive domain benchmarks across different embedding tasks available to evaluate the performance on domain-specific tasks. This is also one of the motivations behind this work. With FinMTEB, we can better evaluate the current domain embedding models.
>
> ---
> ```
> A recent study [2] introducing more in-domain MTEB data (e.g., classification data) has been shown to significantly improve MTEB performance.
> ```
>
> We appreciate the idea of ICL in LLM-based embedding models, confirming that these models are more powerful than previous dense retrievers. This also motivated the creation of our work. We also tested the 'bge-en-icl' model in our work.
>
> Except for the mentioned difference in the '1. Clarification on Domain', in the above study, the performance of general tasks using ICL, such as wiki, web, and news, also improved, only confirming the effectiveness of the ICL approach. They did not compare the differences between different domains. Additionally, we cannot directly compare their results across domains due to variations in datasets, such as sentence complexity.
>
> In our paper, we focused specifically on domain comparison. We compared benchmark performance across different domains while controlling for text complexity and concentrating on domain patterns. We found that under the same text complexity, especially at higher complexity levels, the performance of domain-specific tasks was worse. We believe that the conclusion drawn from this comparison is more robust.
>
> ---
> > 2.  In Tables 8, 9, 10, 11, 12, 13, and 14 of Appendix D, many of the datasets used in this work are in Chinese. Could you clarify this?
>
> These Chinese datasets are not used in the experiments. They are included in the benchmark to provide more options for future research. In the experiments conducted for this study, only the English datasets shown in Figure 1 were used.
>
> For example, the MTEB benchmark includes many different languages, but only English datasets were used in the experiments of the paper.
>
> ---
> We would also like to clarify, as reviewer AfJc and reviewer xpNJ mentioned, that the FinMTEB proposed in this work is a 'valuable tool' and a 'helpful resource.' We hope to pave the way for future research in domain embedding.
>
> Once again, we sincerely thank you for your valuable feedback and remain available for further discussion if you have any additional questions or require more clarification!
>
> ---
> [1] Ling, Chen, et al. "Domain Specialization As the Key to Make Large Language Models Disruptive: A Comprehensive Survey." arXiv preprint arXiv:2305.18703.
>
> [2] Liang Wang, Nan Yang, Xiaolong Huang, Linjun Yang, Rangan Majumder, and Furu Wei. Improving text embeddings with large language models. arXiv preprint arXiv:2401.00368, 2023.

---

> > ### Author Response · Authors · 2024-12-02
> > **Any further questions?**
> >
> > Dear Reviewer DoPN,
> >
> > As we approach the final day of the discussion phase, we want to address any misunderstandings raised regarding our paper. We believe our work is a valuable resource for benchmarking domain embedding models across 64 datasets and paves the way for domain embedding tasks.
> >
> > We hope our clarifications on domain definitions, the generality performance of pre-trained models, and the use of English datasets have been helpful.
> >
> >  If you have any further questions, we are happy to address them!
> >
> > Best, Authors

---

> > > ### Comment · Reviewer_DoPN · 2024-12-02
> > > **Response to Authors**
> > >
> > > Thank you for the detailed response. All my concerns have been addressed, and I have increased my score accordingly.

---

> > > > ### Author Response · Authors · 2024-12-02
> > > > **Thank You!**
> > > >
> > > > Dear Reviewer DoPN,
> > > >
> > > > Thank you for your careful consideration of our work and insightful comments. We appreciate the willingness to raise the score to 6.
> > > >
> > > > Thank you once again, and we wish you all the best!
> > > >
> > > > Best，
> > > >
> > > > The Authors

---

### Author Response · Authors · 2024-11-24
**Summary of Revision**

We sincerely thank the reviewers for their detailed review, valuable feedback, and acknowledgment of the novelty of our work. To address the concerns raised and enhance the quality of our paper, we have made the following revisions:


**Table:**

| Change                             | Section    | Related Reviewers          |
|------------------------------------|------------|----------------------------|
| Revise Title     | - | Reviewer WW39,AfJc             |
| Increase Inter-dataset Semantic Similarity Testing Samples | Appendix A | Reviewer DoPN|
| Statement About Semantic Diversity   | Section 3.2, line 258 | Reviewer WW39|
| Additional SoTA Models   | Appendix B | Reviewer AfJc|
| Experiment on Domain Finetuned Embedding Model   | Appendix H | Reviewer WW39|

---

**Details:**

1. **Revise Title**

Inspired by the comments, we have revised our title to "Do We Need Domain-Specific Embedding Models? An Empirical Investigation on the Finance Domain and a Domain-MTEB Benchmark," which better captures our contribution.

1. **Expanded Data Sample for Inter-dataset Semantic Similarity (Appendix A)**

For robust evaluation, we incresea the data samples from 100 to 1000, and update the corresponding heatmap in Appendix A.

1. **Changed Statement About Semantic Diversity (Section 3.2, line 258)**

* 'most datasets in FinMTEB have an inter-dataset similarity score below 0.6, indicating high semantic diversity'
* to 'most datasets in FinMTEB have an inter-dataset similarity score below 0.6, with a mean cosine similarity of 0.4'

1. **Test Additional SoTA Models (Appendix B)**

We tested the performance of Echo Embedding (Springer et al., 2024) , AnglE-BERT (Li & Li, 2023),NV-Embed v2 (Lee et al., 2024), and BeLLM (Li & Li, 2024). The results are shown in Appendix B along with their evaluation times (batch size = 512).

5. **Supplementary Experiment on Domain Finetuned Embedding Model (Appendix H)**

To demonstrate the reliability of our statements, we fine-tuned e5-mistral-7b-instruct (Wang et al., 2023) using a domain corpus. The results are consistent with our findings, and we will also open-source this model along with our FinMTEB benchmark for the community.


As the discussion period is ending soon, we would greatly appreciate it if you could go through our responses and share any feedback you may have. Once again, thank you very much for reviewing our work! We remain open to providing further results or clarifications to address any additional concerns.

---

### Meta-Review · Area_Chair_WyLr · 2024-12-20

**Metareview:**

The paper asks an intuitive question "Do We Need Domain-Specific Embedding Models?" and performed a case study in the financial domain, where the authors created a dataset.

Reviewers generally gave borderline or rejection scores. The biggest concern is an overclaim from one specific domain to a general statement of their finding. In addition, the question researched does not differ drastically from previous studies of (L)LM domain adaptation. Of course, the authors can argue that the domain adaptation studies focus on different models/aspects, the paper has not offered insights that are interesting enough.

**Additional Comments On Reviewer Discussion:**

Reviewers generally gave borderline or rejection scores. Reviewers who gave borderline leaning positive scores also raised important concerns, such as overclaiming.

---

### Decision · Program_Chairs · 2025-01-22

Reject